# Understanding vs. Generation: Navigating Optimization Dilemma in Multimodal Models

**Sen Ye**[1,2]   **Mengde Xu**[2]   **Shuyang Gu**[2†]   **Di He**[1]   **Liwei Wang**[1,3,4†]   **Han Hu**[2]

[1] State Key Laboratory of General Artificial Intelligence, Peking University   [2] Tencent
[3] Center for Data Science, Peking University  [4] Center for Machine Learning Research, Peking University

## Abstract

Current research in multimodal models faces a key challenge where enhancing generative capabilities often comes at the expense of understanding, and vice versa. We analyzed this trade-off and identify the primary cause might be the potential conflict between generation and understanding, which creates a competitive dynamic within the model. To address this, we propose the Reason-Reflect-Refine (R3) framework. This innovative algorithm re-frames the single-step generation task into a multi-step process of "generate-understand-regenerate". By explicitly leveraging the model's understanding capability during generation, we successfully mitigate the optimization dilemma, achieved stronger generation results and improved understanding ability which are related to the generation process. This offers valuable insights for designing next-generation unified multimodal models. Code is available at https://github.com/sen-ye/R3.

## 1 Introduction

For decades, the pursuit of Artificial General Intelligence (AGI) has been a central goal in AI research. A key step toward this vision is the development of unified multimodal models capable of both understanding and generating visual information, much like humans do (Team et al., 2023; OpenAI, 2025). However, recent advance in large-scale multimodal learning points to a core dilemma that understanding and generation are hard to improve simultaneously (Henighan et al., 2020; Wang et al., 2024; Wu et al., 2025b; Zhang et al., 2025a). For instance, models fine-tuned for high-fidelity image synthesis, such as those based on diffusion architectures, often struggle with tasks requiring precise visual understanding, including object counting or spatial reasoning (Cho et al., 2023; Ghosh et al., 2023; Wei et al., 2025). Conversely, models optimized for tasks like visual question answering (VQA) or dense captioning tend to exhibit weaker creative and generative performance compared to their generative counterparts (Dong et al., 2023; Xie et al., 2024; Wu et al., 2025b).

A number of efforts have been made to resolve this tension. Some researchers argue that different tasks require specialized tokenizers (Wu et al., 2025b; Chen et al., 2025), and thus propose unified tokenization schemes (Team, 2024; Ma et al., 2025) to harmonize representation across modalities. Others attempt to disentangle understanding and generation by designing novel architectures that allocate separate capacity to each function (Liang et al., 2024; Deng et al., 2025; Zhang et al., 2025a). While these approaches achieve partial success, we argue that the crux of the conflict comes from the different training objectives. The generative objective typically maximizes the likelihood of samples under the data distribution, a goal that can be optimized without the understanding ability. As a result, the model's capacity may be monopolized, in competition with the robust understanding required. This raises a fundamental question: should generation actively incorporate the model's understanding of the underlying semantics?

In this paper, we introduce the Reason-Reflect-Refine (R3) framework, which re-conceptualizes generation as a multi-step process rather than a single-shot mapping. Instead of treating generation and understanding as competing objectives, R3 explicitly integrates understanding into the generative loop. By first reasoning the user request and producing an initial draft, then reflecting whether this generated result meets the user request, and finally refining the output according to the reflection,

---

[†]Corresponding Author.

the model transforms understanding from a passive evaluation task into an active component of generation.

Analogous to a painter's creative process, our framework unfolds in distinct stages. In the Reason stage, the model first analyzes the user's intent to conceptualize the final image. It enriches the initial prompt by imagining and incorporating various fine-grained details, producing an explicit textual blueprint before synthesizing an initial draft. Recognizing that high-fidelity generation from complex prompts is rarely achievable in a single attempt, the framework then enters an iterative Reflect–Refine loop. Here, the model evaluates its output against the original prompt—a process that demands strong multimodal understanding. If the output aligns well, the procedure terminates; otherwise, the model formulates corrective textual instructions and refines the image accordingly. This self-correction cycle continues until satisfactory alignment is reached, with the model itself deciding when to stop. The entire process is trained end-to-end using an outcome-based reward signal derived from the final image quality. To further enhance efficiency, we introduce an immediate rollout strategy that accelerates convergence without compromising performance.

Our experiments demonstrate that incorporating reflecting and refining into the generative process enables the model to effectively leverage its understanding capability, leading to substantial improvements across multiple text-to-image benchmarks. Moreover, embedding reflection within generation not only enhances output quality but also exercises the model's understanding ability, thereby preventing the degradation typically caused by modeling capacity competition. On tasks aligned with generative content, we even observe notable gains in understanding—for instance, counting accuracy improves from 79.3 to 84.6. Overall, the proposed R3 framework reconciles the long-standing conflict between generation and understanding: it achieves stronger generation while simultaneously preserving, rather than diminishing, the understanding ability. This establishes a promising path forward for the development of future unified multimodal models.

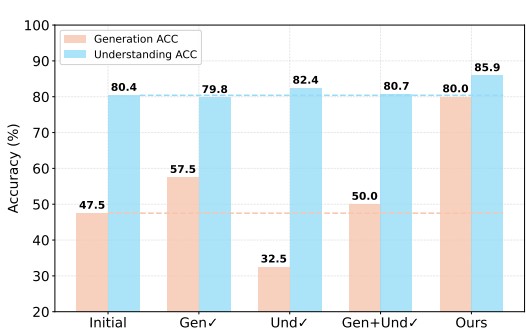

Figure 1: Fine-tuning BAGEL exclusively on generation or understanding degrades the complementary capability. Naive co-training shows minor gains, whereas our proposed method demonstrates significant improvement in both. Results are reported on counting subset of GenEval++.

Our contributions can be summarized as follows:

- We provide a systematic analysis of the conflict between generation and understanding in multimodal large models, and identify its root cause: traditional approaches treat the two as independent tasks, which compete for model capacity and lead to a trade-off where one improves at the expense of the other.

- Building on this insight, we propose the Reason–Reflect–Refine (R3) framework, which decomposes generation into a structured generate–understand–regenerate process. By explicitly incorporating understanding into the generative pipeline, R3 mitigates the conflict caused by separate optimization, yielding stronger generation while preserving understanding ability.

- Extensive experiments verify the effectiveness of R3: by fully leveraging the model's own understanding ability during generation, R3 not only achieves superior generative performance but also avoids the degradation of understanding. These findings shed light on the design of future unified multimodal models and point to new data strategies for balancing generation and understanding.

## 2 METHODOLOGY

### 2.1 UNIFYING GENERATION AND UNDERSTANDING

A key challenge in developing powerful, unified multi-modal models lies in effectively integrating both generation and understanding capabilities. As shown in fig. 1, fine-tuning a model exclusively on one of these tasks degrades its performance on the complementary task. This suggests a competitive

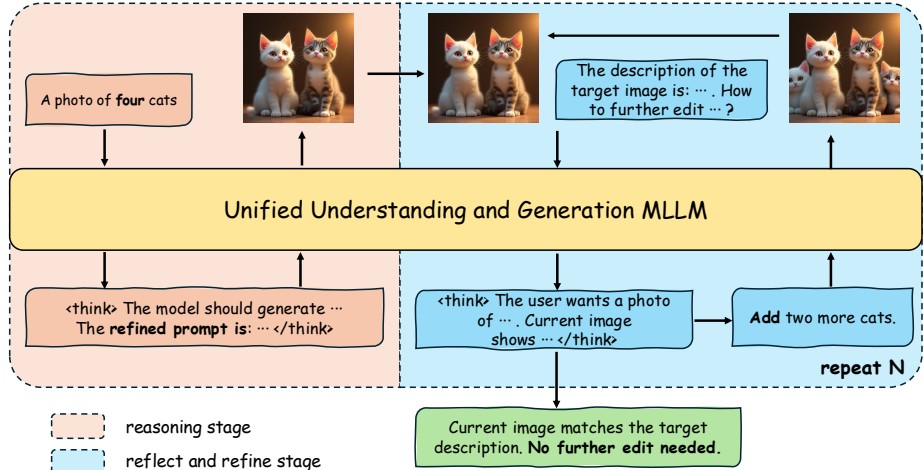

Figure 2: The inference pipeline of our Reason-Reflect-Refine framework. The model starts by Reasoning to produce an initial plan and image. It then enters an iterative Reflect-Refine loop, assessing its output and making corrections until the image aligns with the user's prompt or a stopping condition is met.

relationship between the two tasks, where shared model parameters are optimized for one objective at the expense of the other. Furthermore, naive training on a mixture of data provides only negligible performance gains. This observation supports our core hypothesis: the optimization trajectories for generation and understanding are not inherently aligned. The two tasks are not coupled tightly enough to benefit from mutual training, suggesting they are driven by fundamentally different optimization dynamics. This observation leads us to a crucial question: Is it possible to align the optimization goals of generation and understanding, thereby reducing the conflicts that emerge when training a single model on both?

A possible solution is to intrinsically embedding visual understanding into the generative process. This paradigm shift ensures that the improvement of a model's generation capability is inherently dependent on its understanding ability, thereby preventing the common issue where training for one task degrades the performance of the other. To realize this vision, we introduce the Reason-Reflect-Refine (R3) framework( fig. 2), a novel approach that recasts image generation as an iterative process of reasoning, reflection, and refinement. By embedding image understanding as a core component of a generative chain-of-thought, we enable the model to critically assess and progressively improve its own output, fostering a synergistic relationship between understanding and generation.

## 2.2 FRAMEWORK OVERVIEW

We build our framework on top of a unified multimodal model, BAGEL (Deng et al., 2025) (parametrized by $\theta$), which is capable of image understanding, generation, and editing. The generation process is formulated as a sequence of alternating text and image generation steps:

$$t^1, I^1, \ldots, t^n, I^n \sim \pi_\theta(\cdot|c) \tag{1}$$

where $c$ is the initial user-provided prompt, and at each step, the model generates text $t^i$ auto-regressively or an image $I^i$ by progressive denoising.

To make this process computationally tractable and modular, we decompose it into three distinct, alternating tasks. Assuming a Markovian property, we model the generation as a sequence of these specialized tasks:

1. **Reason**: The process begins with the model expanding on the input prompt $c$ to generate a more detailed, reasoned plan $t^1$. The plan is expected in the format *"<think>plan</think>"*. Then, the model synthesizes an initial image $I^1$ according to this plan. This is modeled by the joint probability $\pi_\theta(I^1, t^1|c) = \pi_\theta(I^1|t^1, c)\pi_\theta(t^1|c)$. While $\pi_\theta(t^1|c)$ is a standard language

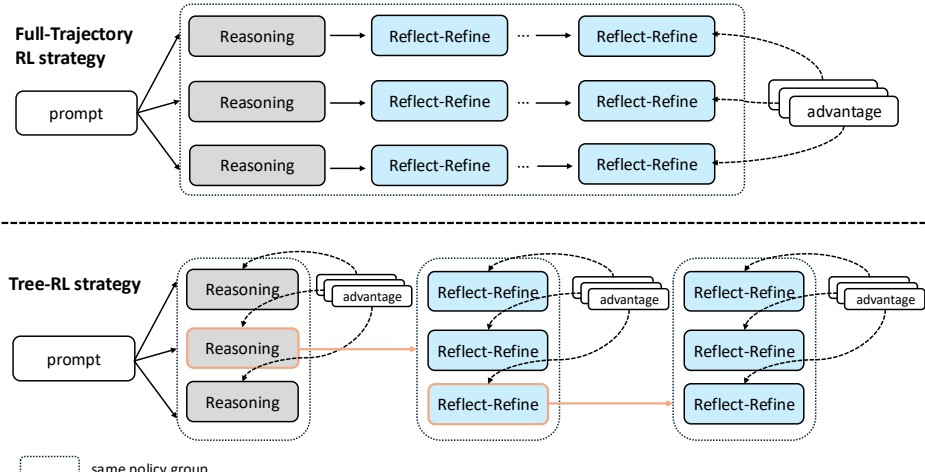

Figure 3: The training procedure, which alternates between optimizing the Reason policy and the Reflect-Refine policies. The replay buffer, populated by the Reason stage, provides on-policy data for training the subsequent stages.

modeling policy, and the diffusion policy $\pi_\theta(I^1|t^1, c)$ can be calculated using stochastic differential equations (SDEs), similar to (Ma et al., 2024; Liu et al., 2025).

2. **Reflect**: Upon obtaining an initial generated image, the model is then required to assess its alignment with the user's original intent c. This critical self-assessment process is termed reflection, which can be formally expressed as $\pi_\theta(t^{i+1}|I^i, c)$. In cases where the generated output is deemed satisfactory, the model is designed to produce a definitive termination signal: "No further edit needed." However, if the output is still deficient, the model performs a critical introspection, identifying the discrepancies between the current image and the desired objective. This introspection culminates in the generation of a refined editing instruction $e^{i+1}$, which serves as guidance for subsequent iterative improvements. To ensure a structured and consistent output, we employ a system prompt that strictly enforces the format: *"<think>reflection</think>editing instruction."*

3. **Refine**: Subsequently, the model executes the generated editing instruction $e^{i+1}$ to modify the previously created image $I^i$, and produce a refined output $I^{i+1}$. This refinement step is formally modeled as a conditional generation process: $\pi_\theta(I^{i+1}|e^{i+1}, I^i)$. The entire reflection-and-refinement loop is performed iteratively, forming a chain-of-thought that continues until the model's internal assessment confirms that the generated image satisfies all aspects of the user's request.

## 2.3 TREE-RL STRATEGY

The full, multi-turn trajectory can be conceptualized as a sequential process: *Reason -> Reflect -> Refine -> Reflect -> Refine....* This sequence can be theoretically framed as the chain-of-thought process of image generation, which could be directly optimized using reinforcement learning (RL). However, training end-to-end reinforcement learning (RL) models presents several challenges. First, the large number of training iterations can lead to error accumulation, causing instability. Second, the lack of explicit supervision for intermediate steps results in low training efficiency. To overcome these issues, we propose a tree-based RL strategy that provides clear supervision for the outcome of each intermediate step.

As illustrated in fig. 3, we split the trajectory into Reason stage and Reflect-Refine stages. Each stage populates its result (result image $I^i$ and current reward) to the next stage as initial condition. To enhance training efficiency and speed up convergence, we employ an importance sampling strategy when selecting for previous stage's results: Sampling more samples with diverse rewards. This strategic focus on error correction allows the model to learn and improve effectively, without

compromising its overall task completion capability. All policies are optimized with the GRPO loss function, as described in section A.2. As shown in fig. 4, by employing this tree-based rollout strategy, our model can be trained effectively on long, complex generation chains, allowing for progressive improvement through multiple rounds of refinement during inference.

## 2.4 STAGE-WISE REWARD

We employ different reward models to evaluate the various stages of our reinforcement learning (RL) training. Specifically, as described in section 2.2, the reasoning stage involves two policy processes: $\pi_\theta(I^1|t^1, c)$ and $\pi_\theta(t^1|c)$.

First, we measure the quality of the initial generated image $I^1$, using a pre-defined reward model $\mathcal{V}$ (a pre-trained Vision-Language Model). This model outputs a scalar score $V_j = \mathcal{V}(I_j^1, c) \in [0, 1]$, which represents the alignment between the image and the prompt. We then define the rewards for each policy as follows:

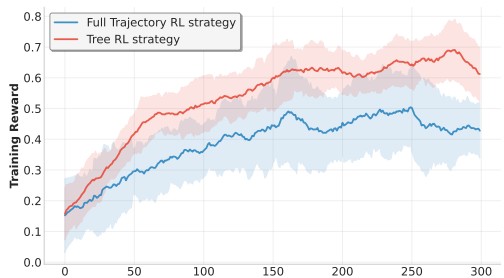

Figure 4: Training reward curves of the Tree-RL versus Full Trajectory RL strategies. The reward curve for Full Trajectory RL is substantially lower than that of Tree-RL. This performance gap is attributed to the high variance and noise introduced by the long trajectories inherent in the full trajectory approach, which complicates the advantage assignment problem.

- For the diffusion generation stage $\pi_\theta(I^1|t^1, c)$, the reward is defined as $r_{j,\text{diffusion}} = V_j$.

- For the text generation stage $\pi_\theta(t^1|c)$, the reward is $r_{j,\text{text}} = V_j + r_{j,\text{format}}$, where $r_{j,\text{format}}$ measures the accuracy of the generated format.

For the Reflect-Refine stage, the model generates a textual reflection based on the image $\hat{I}$ and its reward $\hat{V}$ from the previous step. This process leads to one of two distinct outcomes:

1. **Correction Required**: If the image does not align well with the prompt $c$ (i.e., $\hat{V} < 1$), the model outputs a precise editing instruction $e_j$. The Refine step then generates an improved image $I_j$, which is evaluated to get a new image reward $V_j$.

2. **Correction Unnecessary**: If the image already satisfies the prompt (i.e., $\hat{V} = 1$), the model must output the specific phrase "No further edit needed." This serves as a termination signal for the process.

To encourage meaningful reflection and accurate termination, we use a correctness metric to evaluate the model's output. For an image with a reward of $\hat{V}$, the metric is defined as:

$$\text{C}_j = \begin{cases} V_j - \hat{V} & \text{if } \hat{V} < 1 \\ \mathbb{I}(e_j = \text{"No further edit needed"}) & \text{if } \hat{V} = 1 \end{cases} \tag{2}$$

where $\mathbb{I}(\cdot)$ is the indicator function. This metric is designed to reward two key behaviors: measurable improvement for flawed images (where $V_j > \hat{V}$), and correct termination for images that already meet the prompt's criteria.

The rewards for the reflection and refinement steps are then based on this metric:

$$r_{j,\text{reflection}} = \text{C}_j + r_{j,\text{format}}, \quad r_{j,\text{refinement}} = \text{C}_j \tag{3}$$

While our reinforcement learning objective does not directly optimize understanding tasks, the model develops robust visual comprehension by being trained to evaluate image-prompt alignment via the reflection reward.

# 3 EXPERIMENTS

## 3.1 CO-EVOLUTION OF UNDERSTANDING AND GENERATION

To validate the co-evolution of understanding and generation capabilities within our framework, we conduct controlled experiments on the GenEval++ benchmark (Ye et al., 2025).

**Experimental Setup:** Our training prompts are generated by randomly combining elements from the official templates, ensuring no overlap with the test set. We employ Qwen-2.5-VL-72B as our reward model. By default, our method includes an initial reasoning stage followed by four reflection-refinement stages. Following the original benchmark setting, we use GPT-4.1 to evaluate generation quality.

**Evaluation of Understanding:** To assess the model's understanding capabilities, we introduce two novel evaluation protocols: **Compositional Visual Question Answering (VQA)** and **Image-Text Alignment (ITA)**. The VQA task probes the model's ability to perceive compositional elements in its generated images. The ITA task evaluates its capacity to assess the overall quality and prompt-alignment of the generated images.

For both tasks, we followed a standardized protocol. We first generated a corpus of images using a powerful, pretrained text-to-image generation model. Subsequently, we used the Gemini-2.5-Flash model to create the gold-standard ground truth for both VQA questions and ITA assessments and conduct a careful human alignment check to ensure the quality of annotation. Finally, we evaluated our model's performance by calculating the similarity between its output and the annotated ground truth labels. This rigorous evaluation framework allows us to directly measure the effectiveness of our approach. We provide detailed description in section A.3.

Table 1: Instruction-following generation ability on the GenEval++ benchmark, evaluated by GPT-4.1. Bold indicates the best result. †indicates our framework with only the reasoning stage. Green arrows indicate improvement over the BAGEL baseline.

| Method | Color | Count | Color/Count | Color/Pos | Pos/Count | Pos/Size | Multi-Count | Overall |
|---|---|---|---|---|---|---|---|---|
| GPT-4o (OpenAI, 2025) | 0.900 | 0.675 | 0.725 | 0.625 | 0.600 | 0.800 | 0.850 | 0.739 |
| FLUX.1-Kontext (Labs, 2024) | 0.425 | 0.500 | 0.200 | 0.250 | 0.300 | 0.400 | 0.325 | 0.343 |
| FLUX.1-dev (Labs, 2024) | 0.350 | 0.625 | 0.150 | 0.275 | 0.200 | 0.375 | 0.225 | 0.314 |
| Janus-Pro (Chen et al., 2025) | 0.450 | 0.300 | 0.125 | 0.300 | 0.075 | 0.350 | 0.125 | 0.246 |
| T2I-R1 (Jiang et al., 2025) | 0.675 | 0.325 | 0.200 | 0.350 | 0.075 | 0.250 | 0.300 | 0.311 |
| Echo-4o (Ye et al., 2025) | **0.800** | 0.575 | 0.550 | **0.775** | 0.625 | **0.800** | 0.625 | 0.679 |
| BAGEL (Deng et al., 2025) | 0.325 | 0.600 | 0.250 | 0.325 | 0.250 | 0.475 | 0.375 | 0.371 |
| BAGEL + Ours† | 0.500 | 0.650 | 0.600 | 0.650 | 0.550 | 0.600 | 0.600 | 0.593 ↑0.22 |
| BAGEL + Ours | 0.675 | **0.725** | **0.575** | 0.725 | **0.750** | 0.575 | **0.800** | **0.689** ↑0.32 |

Table 2: Evaluation of understanding capabilities on our proposed ITA benchmarks. All scores are reported as accuracy (%). †indicates our framework with only the reasoning stage. Green arrows indicate improvement over the BAGEL baseline.

| ITA | Color | Count | Color/Count | Color/Pos | Pos/Count | Pos/Size | Multi-Count | Overall |
|---|---|---|---|---|---|---|---|---|
| BAGEL | 60.63 | 58.54 | 45.42 | 63.54 | 53.96 | 80.83 | 61.50 | 60.60 |
| BAGEL + Ours† | 60.42 | 59.38 | 47.71 | 63.75 | 55.63 | 81.46 | 63.96 | 61.76 ↑1.16 |
| BAGEL + Ours | **69.58** | **67.50** | **69.79** | **72.29** | **76.04** | **83.33** | **75.00** | **73.37** ↑12.77 |

The results in table 1 confirm the enhanced generation capabilities of our framework. Even compared with the SOTA method Echo-4o (Ye et al., 2025) which finetunes on related data, our method still achieves 1% point overall improvement. And in complex situation like *Multi-Count*, our method is significantly better then Echo-4o.

As shown in table 2 and table 3, our approach also yields substantial gains in the ITA and VQA understanding tasks. Notably, the reflection-refinement process is critical. While a baseline with only the reasoning stage improves GenEval++ scores, our full framework adds a further improvement of nearly 10% points. This effect is more dramatic in understanding tasks, where the reasoning-only

Table 3: Evaluation of VQA capabilities. All scores are reported as accuracy (%). †indicates our framework with only the reasoning stage. Green arrows indicate improvement over the BAGEL baseline.

| VQA | Color | Count | Color/Count | Color/Pos | Pos/Count | Pos/Size | Multi-Count | Overall |
|---|---|---|---|---|---|---|---|---|
| BAGEL | 91.74 | 79.30 | 88.28 | 77.99 | 82.93 | 85.10 | 92.45 | 86.48 |
| BAGEL + Ours† | 91.67 | 76.12 | 88.76 | 78.71 | 83.29 | 84.98 | 93.45 | 86.72 ↑0.24 |
| BAGEL + Ours | **93.95** | **84.63** | **91.15** | **84.09** | **86.06** | **86.54** | **94.50** | **89.63** ↑3.15 |

| One frog jumped in front of three suitcases, showcasing a playful moment in the scene. | Three ducks quacked by the pond, all gathered at the side, basking in the serene water's reflection. | Evidence of a currently present invisible elephant, such as large footprints in the mud, broken branches, and disturbed water in a pond | A picture of a birthday card featuring a colorful mermaid tail design, with the text on it: "HAPPY", "BIRTHDAY", "OUR", "MERMAID". | A miniature diorama scene using tilt-shift photography technique, depicting a chibi-style scene of "Sun Wukong's battle with the White Bone Spirit" |
|---|---|---|---|---|

Figure 5: Qualitative comparison between Bagel and our results.

baseline offers minimal gains (1.16 on ITA, 0.24 on VQA), whereas our full framework achieves significant improvements (12.77 on ITA, 3.15 on VQA). These findings highlight that the reflection-refinement stage is essential for improving generation and is the key to unlocking the model's understanding abilities. We also provide some qualitative results in fig. 5. For a more detailed visualization of the multi-round generation process, please refer to the section A.6.

## 3.2 ABLATION STUDIES

**Influence of Trajectory Length.** We investigate the impact of trajectory length on performance during both training and inference. For training, as detailed in table 4, a trajectory length of two (one reasoning stage and one reflection-refinement stage) achieves an optimal balance between computational cost and performance.

For our inference-time analysis, we evaluate the impact of trajectory length by setting a maximum number of reflection-refinement turns. For each sample, the evaluation terminates in one of two ways:

Table 4: Impact of training trajectory length on generation (GenEval++ Validation Reward) and understanding (ITA). "RR" denotes a Reflection-Refinement stage.

| Rollout Stage | GenEval++ | ITA |
|---|---|---|
| Reason | 0.654 | 62.83 |
| Reason + $1\times$ RR | 0.729 | 74.49 |
| Reason + $2\times$ RR | 0.732 | 74.76 |

Table 5: Cross-topic evaluation on GenEval++. Models are trained on specific category and tested on all categories.

| Train \ Test | Counting | Color | Position/Size |
|---|---|---|---|
| Pretrained | 62.71 | 60.63 | 80.83 |
| Counting | **71.25** | 60.63 | 81.46 |
| Color | 62.50 | **66.25** | 81.25 |
| Position/Size | 60.00 | 60.00 | **83.33** |

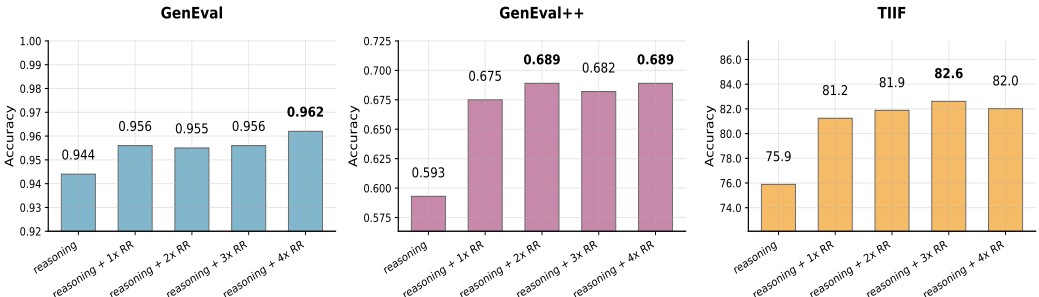

Figure 6: Inference-time scaling effect across the GenEval, GenEval++, and TIIF benchmarks (left to right). Performance is shown as a function of the maximum allowed reflection-refinement turns.

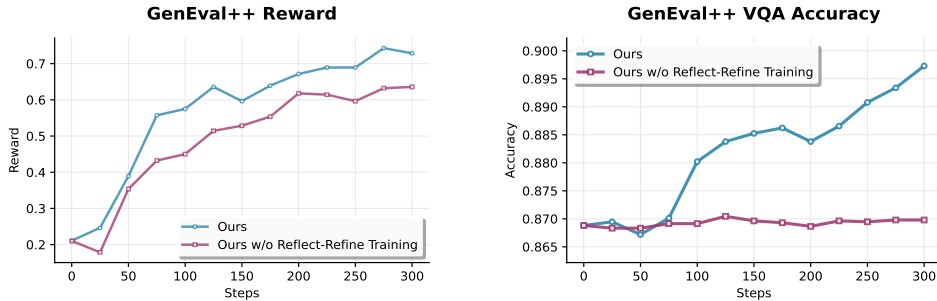

Figure 7: The evolution of generation and understanding abilities in the training process. The left figure shows the generation accuracy (measured by Qwen-2.5-VL-72B.) and the right figure shows the VQA accuracy.

either the model signals completion by outputting "No further edit needed," or it is stopped upon reaching the predefined maximum number of turns. As illustrated in fig. 6, the results across the GenEval (Ghosh et al., 2023), GenEval++ (Ye et al., 2025), and TIIF (Wei et al., 2025) benchmarks show a consistent pattern. The most substantial performance gain occurs after the first reflection-refinement turn. While further improvements are observed with additional turns, the performance generally saturates, reaching its peak when the trajectory length is extended to four or five turns.

**Capabilities Evolution During Training.** fig. 7 demonstrates how our proposed framework drives the simultaneous improvement of generation and understanding capabilities. In the first 150 steps, our model's generation accuracy follows a trajectory similar to the reflection-free baseline, while the VQA accuracy shows minimal change. This suggests that the initial training primarily focuses on basic generative mapping without deepening internal understanding. However, beyond the 150-step mark, the reflection-refinement mechanism begins to yield observable results. At this point, the model's understanding ability starts to rise noticeably. This gain in understanding does not occur in isolation; it is directly associated with a subsequent acceleration in generation accuracy, allowing our model to achieve superior performance relative to the baseline. Ultimately, the comprehensive results validate the effectiveness of our framework in mitigating the optimization dilemma and demonstrating that explicitly integrated understanding is key to unlocking superior, unified multimodal performance.

**Generalization of co-evolutionary Learning.** A key question is whether the co-evolution of understanding and generation capabilities generalizes to broader domains beyond GenEval++. While rigorously measuring this co-evolution would require specialized evaluation protocols, as we constructed for GenEval++, we can assess the impact on general generation capabilities. Our results on the TIIF benchmark (table 6) demonstrate that our framework yields significant improvements in a general-domain setting, suggesting that the benefits of our approach do transfer.

To further probe the nature of the learned understanding, we conducted a cross-topic experiment within GenEval++. We trained our model on specific attribute categories (e.g., color, counting) and evaluated its understanding on all categories. The results, presented in table 5, indicate that the improvements in understanding are localized to the domains the model was trained on. This finding suggests that our current framework learns domain-specific understanding. While this specialized

Table 6: **Quantitative evaluation results on TIIF testmini benchmark (Wei et al., 2025).** Results are evaluated by QwenVL2.5-72B. * indicates our reproduced results using the official repository and checkpoint. The best results are **bolded**.

| Model | Overall | Basic Following | | | | Advanced Following | | | | | | Designer |
|---|---|---|---|---|---|---|---|---|---|---|---|---|
| | | Avg | Attribute | Relation | Reasoning | Avg | Attribute +Relation | Attribute +Reasoning | Relation +Reasoning | Style | Text | |
| GPT-4o (OpenAI, 2025) | 84.19 | 85.30 | 81.00 | 86.16 | 88.74 | 81.24 | 81.95 | 80.03 | 80.88 | 76.67 | 92.76 | 89.55 |
| FLUX.1-dev (Labs, 2024) | 66.24 | 74.41 | 72.50 | 78.20 | 72.52 | 60.72 | 66.76 | 61.76 | 56.60 | 63.33 | 44.49 | 74.63 |
| FLUX.1-Pro (Labs, 2024) | 63.75 | 71.39 | 70.00 | 68.51 | 75.66 | 64.63 | 70.69 | 62.34 | 64.65 | 63.00 | 34.39 | 69.94 |
| Janus-Pro-7B (Chen et al., 2025) | 65.38 | 74.99 | 74.50 | 73.69 | 76.77 | 61.77 | 65.71 | 62.01 | 61.16 | 43.33 | 38.46 | 79.48 |
| T2I-R1 (Jiang et al., 2025) | 67.61 | 81.14 | 80.50 | **83.09** | 79.81 | 67.38 | 69.92 | 70.10 | 68.69 | 50.00 | 32.13 | 74.25 |
| BAGEL (Deng et al., 2025)* | 70.97 | 78.16 | 78.00 | 80.24 | 76.25 | 68.23 | 73.37 | 64.36 | 68.92 | 80.00 | 40.72 | 76.87 |
| BAGEL *w/* self-CoT (Deng et al., 2025)* | 68.06 | 77.63 | 75.00 | 78.55 | 79.33 | 71.24 | **77.65** | 69.77 | 72.93 | 69.93 | 26.24 | 69.78 |
| BAGEL + Ours | **82.02** | **85.07** | **84.00** | 82.61 | **88.58** | 78.52 | 76.96 | 80.07 | **77.43** | **93.33** | **72.40** | **82.84** |

knowledge is highly effective, developing methods to foster more generalized understanding remains an important direction for future research.

# 4 RELATED WORK

**Unified Large Multi-modal Models.** Unifying and utilizing multi-modal representations is a central goal in the development of large language models. Early work, such as Chameleon (Team, 2024), unified image and text by utilizing discrete tokens for both generation and understanding, employing a unified next-token prediction paradigm. Later works (Wu et al., 2025b; Chen et al., 2025; Wu et al., 2024b; Ma et al., 2025; Tang et al., 2025; Lin et al., 2025; Wang et al., 2024) extended this paradigm with improved tokenizers, decoupled encoders for generation and understanding, and even lossless continuous tokens. In contrast, Wu et al. (2024a) unifies the representation of different modalities in the latent space, utilizing different encoders and decoders for each modality. More recent work (Zhou et al., 2024; Xie et al., 2024; 2025; Deng et al., 2025) has further bridged the modality gap by employing different tasks within the same model: next-token prediction for understanding and a diffusion objective for generation. Among these, BAGEL (Deng et al., 2025) has emerged as a particularly powerful model, demonstrating significantly improved performance on both generation and understanding tasks. In this paper, we also focus on unified multi-modal modeling, but from a task-oriented perspective. Specifically, we aim to unify tasks by framing understanding as a subtask of generation. We adopt BAGEL as our baseline model to ensure a convincing evaluation.

**Reinforcement Learning for Multi-modal Models.** Reinforcement Learning (RL) has emerged as a powerful paradigm for enhancing the reasoning capabilities of large language models (Jaech et al., 2024; Shao et al., 2024). This approach enables models to transcend the limitations of imitation learning, where they merely replicate demonstrated patterns. Instead, RL empowers them to autonomously discover and optimize complex, multi-step generative strategies. Early works Team et al. (2025); Zhou et al. (2025) aimed to adopt the RL paradigm to image understanding, which still focused on text level learning. Recent works have begun to leverage RL to improve text-to-image synthesis (Jiang et al., 2025; Zhang et al., 2025b; Duan et al., 2025; Liu et al., 2025). For instance, GoT-R1 (Duan et al., 2025), built on the next-token prediction paradigm, proposes learning a detailed semantic plan and layout for image generation. T2I-R1 (Jiang et al., 2025) further employs reinforcement learning to jointly train both text and image tokens. FlowGRPO (Liu et al., 2025) extends this approach to diffusion models by applying Generative Reward Process Optimization (GRPO) to the series of denoising steps. Since our work builds on BAGEL, which generates text via discrete next-token prediction and images via a diffusion process, we adopt both GRPO and FlowGRPO to jointly optimize both processes. However, our focus is on recasting understanding as a component of generation. We therefore compose multiple tasks to construct the generation process, resulting in a more complex and thorough framework.

## 5 CONCLUSION

This paper analyzed the key trade-off between generation and understanding in multimodal models, identifying the conflict may arise from their competitive optimization objectives. To navigate this challenge, we proposed the Reason-Reflect-Refine (R3) framework. R3 re-conceptualizes generation as a multi-step process where the model explicitly leverages its understanding capability to iteratively refine its output. Our experiments indicated that by integrating reflection into the generative flow, R3 helps to ease the optimization dilemma, leading to stronger generation results and improved understanding ability related to the generative task. This framework offers valuable insights for designing next-generation unified multimodal models.

## ACKNOWLEDGMENTS

LW is supported by National Science and Technology Major Project (2022ZD0114902) and National Science Foundation of China (NSFC92470123, NSFC62276005).

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

# A APPENDIX

## A.1 CLARIFICATION ON THE USE OF LLM

During the preparation of this manuscript, Large Language Model (LLM) was utilized as a tool to enhance readability and correct grammatical errors. The authors carefully reviewed and edited all AI-generated suggestions to ensure the final text aligns with their original intent and arguments. The intellectual content, analyses, and all arguments presented in this paper are solely the work of the human authors, who take full responsibility for the final content of the publication.

## A.2 RL TRAINING WITH GRPO

Our methodology leverages Group-Relative Policy Optimization (GRPO) to refine two key components: the Chain-of-Thought (CoT) generation within our Reasoning and Reflect stages, and the denoising process for image generation and editing. For the text-based CoT policy, we employ the standard GRPO algorithm (Shao et al., 2024). For the diffusion model, we adopt the framework of FlowGRPO (Liu et al., 2025). Below, we provide a technical overview of these reinforcement learning techniques.

### A.2.1 GROUP-RELATIVE POLICY OPTIMIZATION (GRPO)

Group-Relative Policy Optimization (GRPO) (Shao et al., 2024) is a policy gradient algorithm derived from Proximal Policy Optimization (PPO). It stabilizes training by normalizing advantages against a group of responses sampled from the policy. This approach reduces the variance of advantage estimates, leading to more consistent and effective policy updates.

The optimization process begins by sampling a group of $G$ responses $\{o_i\}_{i=1}^{G}$ for a given prompt $q$ using the current policy $\pi_{\theta_{\text{old}}}$. Each response $o_i$ is evaluated to obtain a reward $R_i$. The group-relative advantage $\hat{A}_{i,t}$ for the $i$-th response is then calculated by standardizing its reward against the statistics of the entire group:

$$\hat{A}_{i,t} = \frac{R_i - \text{mean}\left(\{R_j\}_{j=1}^{G}\right)}{\text{std}\left(\{R_j\}_{j=1}^{G}\right) + \delta},$$

where $\delta$ is a small constant added for numerical stability.

The policy $\pi_\theta$ is then updated by maximizing a clipped surrogate objective. This objective is regularized with a KL-divergence penalty to prevent large deviations from a reference policy $\pi_{\text{ref}}$ (typically the initial supervised fine-tuned model):

$$\mathcal{J}_{\text{GRPO}}(\theta) = \mathbb{E}_{\substack{(q,a)\sim\mathcal{D}, \\ \{o_i\}_{i=1}^{G}\sim\pi_{\theta_{\text{old}}}(\cdot|q)}}$$
$$\left[\frac{1}{\sum_{i=1}^{G}|o_i|}\sum_{i=1}^{G}\sum_{t=1}^{|o_i|}\min\left(r_{i,t}(\theta)\,\hat{A}_{i,t},\,\text{clip}\left(r_{i,t}(\theta),\,1-\varepsilon,\,1+\varepsilon\right)\hat{A}_{i,t}\right) - \beta\,D_{\text{KL}}\left(\pi_\theta\|\pi_{\text{ref}}\right)\right].$$

Here, $r_{i,t}(\theta)$ is the per-token importance sampling ratio between the new policy $\pi_\theta$ and the old policy $\pi_{\theta_{\text{old}}}$:

$$r_{i,t}(\theta) = \frac{\pi_\theta\left(o_{i,t}\mid q,\,o_{i,<t}\right)}{\pi_{\theta_{\text{old}}}\left(o_{i,t}\mid q,\,o_{i,<t}\right)}.$$

### A.2.2 FLOWGRPO FOR DIFFUSION MODEL OPTIMIZATION

To apply policy optimization to our diffusion model, we utilize FlowGRPO (Liu et al., 2025), which adapts the GRPO framework for continuous state-space models trained via flow matching.

**Flow Matching Preliminaries.** Our diffusion model is built upon Rectified Flow (Liu et al., 2022), which defines a linear interpolation between a data sample $x_0 \sim \mathcal{X}_0$ and a noise sample $x_1 \sim \mathcal{X}_1$ as $x_t = (1-t)x_0 + tx_1$ for $t \in [0, 1]$. A velocity field network $v_\theta(x_t, t)$ is trained to predict the vector field $v = x_1 - x_0$ via the flow matching objective:

$$\mathcal{L}_{\text{FM}}(\theta) = \mathbb{E}_{t,x_0,x_1}\left[\|(x_1 - x_0) - v_\theta((1-t)x_0 + tx_1, t)\|_2^2\right].$$

**MDP Formulation and Policy Optimization.** The iterative denoising process of a diffusion model can be naturally framed as a Markov Decision Process (MDP). At each discrete step $t$, the state is $s_t = (c, t, x_t)$, comprising the conditioning prompt $c$, the current time $t$, and the noisy sample $x_t$. The action $a_t$ corresponds to generating the subsequent, less noisy sample $x_{t-1} \sim \pi_\theta(x_{t-1}|x_t, c)$. A terminal reward $R(x_0, c)$ is assigned only at the final step $(t = 0)$ upon generating the complete sample $x_0$.

Under this MDP formulation, FlowGRPO generates a group of $G$ images $\{x_0^i\}_{i=1}^G$ for a given prompt $c$. The advantage for the $i$-th trajectory is computed relative to the group's final rewards:

$$\hat{A}^i = \frac{R(x_0^i, c) - \text{mean}(\{R(x_0^j, c)\}_{j=1}^G)}{\text{std}(\{R(x_0^j, c)\}_{j=1}^G) + \delta}.$$

Since the reward is terminal, the advantage $\hat{A}^i$ is constant across all timesteps $t$ for a given trajectory $i$. The FlowGRPO objective is then:

$$\mathcal{J}_{\text{Flow-GRPO}}(\theta) = \mathbb{E}_{c, \{x^i\}_{i=1}^G}$$
$$\left[ \frac{1}{G} \sum_{i=1}^G \frac{1}{T} \sum_{t=1}^T \min\left(r_t^i(\theta)\hat{A}^i, \text{clip}(r_t^i(\theta), 1-\epsilon, 1+\epsilon)\hat{A}^i\right) - \beta D_{\text{KL}}(\pi_\theta \| \pi_{\text{ref}}) \right],$$

where the importance ratio is $r_t^i(\theta) = \frac{p_\theta(x_{t-1}^i|x_t^i, c)}{p_{\theta_{\text{old}}}(x_{t-1}^i|x_t^i, c)}$.

**Stochastic Differential Equation for Exploration.** Reinforcement learning requires policy exploration, yet the generative process defined by an Ordinary Differential Equation (ODE), $dx_t = v_\theta(x_t, t)dt$, is deterministic. To facilitate exploration, FlowGRPO converts the ODE into a corresponding Stochastic Differential Equation (SDE):

$$dx_t = \left( v_\theta(x_t, t) + \frac{\sigma_t^2}{2t}(x_t + (1-t)v_\theta(x_t, t)) \right) dt + \sigma_t dw_t,$$

where $dw_t$ represents the increments of a Wiener process and $\sigma_t$ controls the noise magnitude. Discretizing this SDE with the Euler-Maruyama method yields the update rule for the generative sampling process:

$$x_{t-\Delta t} = x_t - \left( v_\theta(x_t, t) + \frac{\sigma_t^2}{2t}(x_t + (1-t)v_\theta(x_t, t)) \right) \Delta t + \sigma_t \sqrt{\Delta t} z, \quad z \sim \mathcal{N}(0, I).$$

The noise schedule is given by $\sigma_t = a\sqrt{t/(1-t)}$, where $a$ is a scalar hyperparameter that adjusts the level of stochasticity. This SDE-based sampling enables the policy to explore different generation paths, which is essential for effective RL-based optimization.

**Efficient Training with MixGRPO (Li et al., 2025).** To mitigate the computational demands of SDE-based sampling during policy training, we adopt the MixGRPO (Li et al., 2025) strategy. This approach combines SDE and ODE sampling in denoising process, substantially reducing the computational burden while maintaining strong model performance.

### A.3 EXPERIMENTAL DETAILS

#### A.3.1 GENERATION BENCHMARKS

To validate the effectiveness of our proposed **Reason-Reflect-Refine** framework, we conducted experiments on three widely-used text-to-image generation benchmarks.

**GenEval.** The GenEval benchmark (Ghosh et al., 2023) is designed to assess a model's capability to generate images with complex compositional requirements, including object counting, spatial relationships, and attribute binding. For training, prompts were generated using the official GenEval scripts, which construct prompts by randomly combining predefined templates. We rigorously excluded any prompts present in the official test set from our training data. The reward signal for our reinforcement learning process is derived from the soft reward function proposed by FlowGRPO (Liu et al., 2025), which was specifically designed for this benchmark. The result is presented in table 8, our method reaches a new state-of-the-art with a score of 0.962.

Table 7: Hyperparameters for Different Datasets

| Hyperparams | GenEval | GenEval++ | TIIF |
|---|---|---|---|
| Learning Rate | 5e-6 | 5e-6 | 5e-6 |
| Batch Size | 16 | 16 | 16 |
| Group Size | 16 | 16 | 16 |
| Training Steps | 400 | 300 | 200 |
| CE Weight | 1 | 1 | 1 |
| MSE Weight | 2 | 2 | 2 |
| Temperature | 0.9 | 0.9 | 0.9 |
| Number Timesteps for Reasoning | 10 | 15 | 15 |
| Number Timesteps for Edit | 20 | 20 | 20 |
| CFG for Text | 4 | 4 | 4 |
| CFG for Image | 1.5 | 1.5 | 1.5 |
| KL for Image | 0.005 | 0.005 | 0.005 |
| KL for Text | 0.0005 | 0.0005 | 0.0005 |

**GenEval++.** GenEval++ (Ye et al., 2025) is a more challenging extension of GenEval, featuring more complex instructions and employing advanced Vision-Language Models (VLMs) for a more robust evaluation. Following a similar procedure to GenEval, our training prompts were generated by randomly combining elements from the official templates, ensuring no overlap with the test set. For this benchmark, we employ Qwen-2.5-VL-72B as the reward model.

**TIIF.** The TIIF benchmark (Wei et al., 2025) is designed to systematically assess a model's ability to interpret and follow intricate, fine-grained textual instructions. It comprises 5,000 prompts categorized into three difficulty levels, enabling a nuanced evaluation of critical capabilities such as attribute synthesis, reasoning, and style control. For our experiments, we utilize the official training set prompts for reinforcement learning and report performance on the short-prompt version of the test-mini evaluation set. The reward signal is calculated based on a list of yes/no questions with ground-truth answers provided by the benchmark. The reward $r$ is defined as the ratio of correct answers:

$$r = \frac{N_{\text{correct}}}{N_{\text{total}}}$$

Given model generated image, we utilize Qwen-2.5-VL-72B as the VLM to answer these questions and compute the reward.

### A.3.2 Understanding Benchmarks

A central hypothesis of our work is that structuring the generation process through iterative reasoning with reinforcement learning not only improves the final output but also enhances the model's fundamental multimodal understanding capabilities. To empirically validate this claim, we designed two novel downstream evaluation tasks to measure the change in the Bagel model's comprehension abilities before and after training with our Reason-Reflect-Refine framework. For both tasks, we use a powerful proprietary VLM, Gemini 2.5 Flash, to establish a high-quality ground truth for comparison.

**Compositional Visual Question Answering (VQA).** This task is designed to directly probe the model's ability to accurately perceive and verify the complex compositional elements it was trained to generate, such as object counts, attributes, and spatial relationships.

- **Data Construction.** We first utilize the Bagel model to generate a set of images based on the test prompts from the GenEval++ benchmark. Subsequently, we leverage the structured metadata of these prompts to automatically construct a corresponding set of factual, closed-ended (yes/no) questions. For instance, for a prompt describing "five cats," we generate the question, "Are there exactly 5 cats in the image?".

- **Evaluation Protocol.** Ground truth is established by querying Gemini 2.5 Flash with these image-question pairs and recording its answers. We conduct a careful human alignment check to ensure a high-quality annotation. We then evaluate the accuracy of both the baseline (pre-trained) Bagel

Table 8: **Text-to-image generation ability on the GenEval benchmark (Ghosh et al., 2023).** The best scores are **bolded**.

| Method | Single object | Two object | Counting | Colors | Position | Color attribution | Overall |
|---|---|---|---|---|---|---|---|
| FLUX.1-dev (Labs, 2024) | 0.99 | 0.81 | 0.79 | 0.74 | 0.20 | 0.47 | 0.67 |
| Qwen-Image-RL (Wu et al., 2025a) | 1.00 | 0.95 | 0.93 | 0.92 | 0.87 | 0.83 | 0.91 |
| FlowGRPO (Liu et al., 2025) | 1.00 | 0.99 | 0.95 | 0.92 | **0.99** | 0.86 | 0.95 |
| BAGEL (Deng et al., 2025) | 0.99 | 0.94 | 0.81 | 0.88 | 0.64 | 0.63 | 0.82 |
| BAGEL + Ours | **1.00** | **1.00** | **0.95** | **0.95** | 0.98 | **0.89** | **0.96** |

model and our final (RL-trained) model on this VQA set. The primary metric is the accuracy of each model's answers when compared against the Gemini 2.5 Flash-generated ground truth. We conduct a careful human alignment check to ensure a high-quality annotation.

**Image-Text-Alignment (ITA)**   This task assesses a more holistic and nuanced form of comprehension by evaluating the model's ability to function as a reliable "judge" or reward model, a critical capability for advanced AI systems.

- **Task Setup.** Using the same set of images generated for the VQA task, we prompt the Bagel model to perform an evaluation. Specifically, we employ the official GenEval++ evaluation template, which instructs the model to assess the quality and prompt-alignment of a given image.

- **Evaluation Protocol.** We first obtain a ground-truth set of evaluations by having Gemini 2.5 Flash perform the same assessment on all image-prompt pairs. We conduct a careful human alignment check to ensure a high-quality annotation. We then compare the judgments made by the baseline and the final trained Bagel models against this ground truth. The primary metric is the agreement rate between Bagel's judgments (e.g., its scores or categorical decisions) and the assessments provided by Gemini 2.5 Flash. This allows us to quantify the improvement in the model's ability to discern high-quality, prompt-adherent generations.

### A.3.3   IMPLEMENTATION DETAILS

We detail the specific training procedure of the R3 framework. In the Reasoning stage, we utilize a prompt batch size of 16 and sample 16 responses for each prompt. The reasoning text is sampled with temperature of 0.9, while the diffusion SDE sampling uses a noise parameter of $a = 0.7$. To transition to the Reflect-Refine stage, we select a subset of 16 samples from the total pool of 256 candidates (16 prompts $\times$ 16 responses). Crucially, we employ a sampling strategy to ensure that approximately 20% of these selected instances are "perfect" samples (i.e., achieving a reward of 1). For the subsequent Reflect-Refine stages, we maintain a text sampling temperature of 0.9 but adjust the editing diffusion noise parameter to $a = 1.0$. The training trajectory length is set 2 with 1 round for reasoning and another for reflect-refine. Additional training hyperparameters are summarized in Table 7.

### A.4   EFFECT OF RL TRAINING

To clarify the effect of RL training, we conduct an ablation study comparing Bagel (without RL training) against our RL-trained model using the R3 framework. The results are presented in table 9.

| Inference Strategy | Bagel | Ours |
|---|---|---|
| Reasoning only | 0.399 | 0.593 |
| Reason+RR $\times$ 1 | 0.432 | 0.675 |
| Reason+RR $\times$ 2 | 0.436 | 0.689 |
| Reason+RR $\times$ 3 | 0.439 | 0.682 |
| Reason+RR $\times$ 4 | 0.439 | 0.689 |
| Reason+RR $\times$ 5 | 0.439 | - |

Table 9: Inference performance on GenEval++ comparison between Bagel and R3 (ours) under same inference strategies.

These results demonstrate the effectiveness of RL training:

1. Achieve higher performance ceiling: Our RL-trained model achieves substantially better performance across all iteration counts (0.593 vs. 0.399 with reasoning only, and 0.689 vs. 0.439 at convergence), indicating that RL fundamentally enhances the model's capability to understand and generate improved responses.

2. Improve efficiency: Our model converges to near-optimal performance within 2 Reflection-Refine rounds (0.689), whereas Bagel requires 3 rounds to reach its plateau (0.439). This demonstrates that RL training not only elevates the performance ceiling but also accelerates convergence, making the approach more computationally efficient in practice.

Notably, the performance gap between our model and Bagel increases with the Reflection-Refine framework, suggesting that RL training better equips the model to leverage iterative refinement effectively.

## A.5    COMPUTATIONAL COST ANALYSIS

A key practical concern for iterative refinement is computational overhead. We provide a computational cost analysis as follows:

**Adaptive Inference.**    Unlike fixed-iteration approaches, R3 learns to self-terminate when generation quality is satisfactory. On GenEval++, the distribution of refinement iterations shows efficient resource allocation: 45% of prompts finish immediately (0 refinement cost), 26% require 1 refinement turn, 14% need 2 refinement turns, and only 15% require 3+ refinement turns. This adaptive behavior significantly reduces average computational cost.

**Inference Latency.**    We measure wall-clock time on a single NVIDIA H20 GPU (batch size 1, $512\times512$ resolution). The initial Reasoning stage takes 20–25s. Each Reflect-Refine turn requires 25–35s, with Reflection (text) accounting for 5–10s and Refinement (image) for 20–25s.

## A.6    VISUALIZATION ON MULTI-ROUND GENERATION

We demonstrate the multi-round editing process of our proposed R3 framework in figs. 8 to 15. The figure details each stage, including the initial Reason stage's planning text, the subsequent reflection text, and the resulting refined image.

The boy is on the side of the pond, not under a tree.

<think> The model should generate an image of a boy standing near a pond, not under any tree. The comprehensive prompt is: A young boy stands near the edge of a calm, reflective pond under a clear sky, surrounded by lush greenery. The scene is ultra-realistic, with warm natural lighting and sharp details of the boy's clothing and the textured water surface. </think>

<think> The description states "The boy is on the side of the pond, not under a tree." In the provided image, the boy is standing under a tree near the pond, which contradicts the target description. To align the image with the target description, the tree under which the boy is standing must be removed or replaced with an open area, ensuring the boy appears on the side of the pond without any tree overhead. The instructions should specify editing the tree out of the scene, leaving the boy standing beside the pond in an open environment. </think> Modify the image to show the boy standing directly on the edge of the pond without any trees above him; remove the tree canopy and replace it with an open sky or background, ensuring the boy appears beside the water in an open setting. Final edit: Remove the tree branches and foliage from above the boy, leaving him standing on the pond's edge in an open-air environment with no tree overhead.

<think> The description states that the boy should be "on the side of the pond, not under a tree." In the provided image, the boy is already standing beside the pond, which aligns with the location requirement. There is no tree under which the boy is standing, fulfilling the "not under a tree" condition. Therefore, no further editing operations are necessary to conform to the target description. </think> No further edit needed; the existing image already depicts the boy standing on the side of the pond without being under a tree, matching the description.

Figure 8: An illustration of our model's two-stage generation process. For the prompt "The boy is on the side of the pond, not under a tree," the model's initial reasoning stage erroneously generates a tree above the boy. The subsequent reflect-refine stage corrects this error, after which the process terminates automatically.

A yellow grasshopper hops under the wooden fence as the yellow sun splashes warmth across the field.

<think> The model should generate an image of a yellow grasshopper near a wooden fence, with a bright yellow sun shining over a field. The comprehensive prompt is: A vibrant yellow grasshopper leaps near a rustic wooden fence in a sunlit field, bathed in warm golden light from a glowing yellow sun overhead, ultra-realistic, warm-toned, dynamic lighting. </think>

<think> The description specifies a "yellow grasshopper," but the image shows a green grasshopper. To align with the target description, the grasshopper's color should be edited to yellow while maintaining the rest of the image's elements, such as the wooden fence, warm yellow sun, and field lighting, as they already match the description. </think> Modify the grasshopper to appear yellow while preserving the existing wooden fence, glowing yellow sun, and warm-toned field lighting from the original image. Final edit: Update the grasshopper's color to yellow, ensuring consistency with the target description of a "yellow grasshopper," while retaining all other visual components of the scene as originally presented. Final Output: A yellow grasshopper hops under the wooden fence as the yellow sun bathes the field in warmth, matching the target description.

<think> The description matches the provided image perfectly, as it shows a yellow grasshopper under a wooden fence with a yellow sun casting warmth over a field. No further editing is required to align the image with the target description. </think> No further edit needed, as the existing image already corresponds to the description: "A yellow grasshopper hops under the wooden fence as the yellow sun splashes warmth across the field."

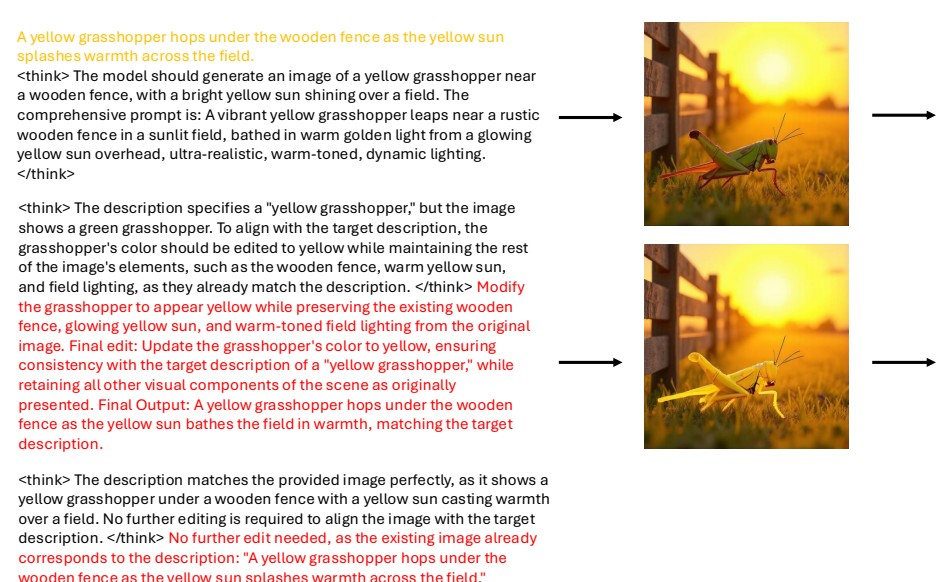

Figure 9: An illustration of our model's two-stage generation process. For the prompt "A yellow grasshopper hops under the wooden fence as the yellow sun splashes warmth across the field," the model's initial reasoning stage generates a grasshopper with an incorrect color. The subsequent reflect-refine stage corrects the color, after which the process terminates automatically.

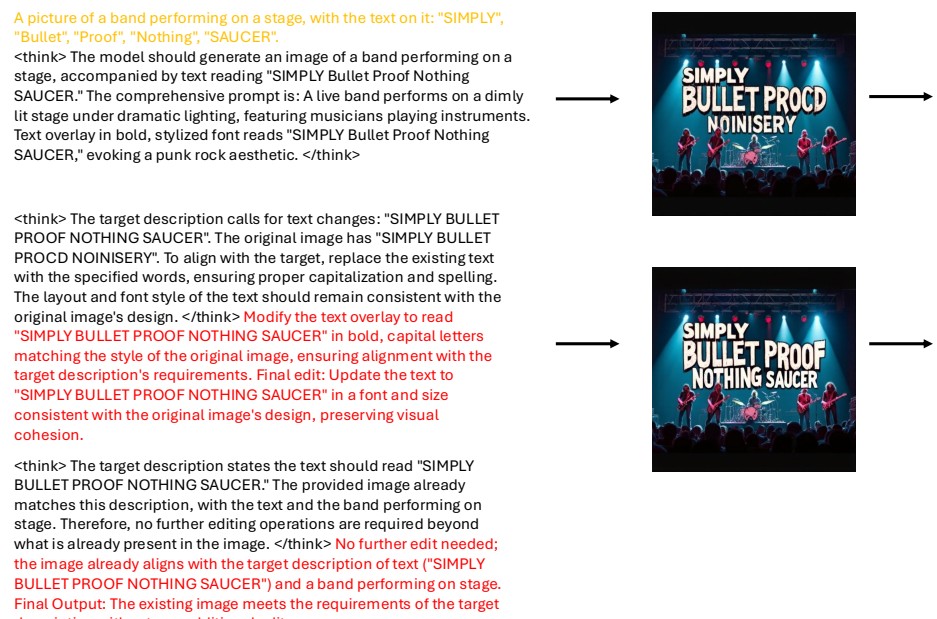

A picture of a band performing on a stage, with the text on it: "SIMPLY", "Bullet", "Proof", "Nothing", "SAUCER".

<think> The model should generate an image of a band performing on a stage, accompanied by text reading "SIMPLY Bullet Proof Nothing SAUCER." The comprehensive prompt is: A live band performs on a dimly lit stage under dramatic lighting, featuring musicians playing instruments. Text overlay in bold, stylized font reads "SIMPLY Bullet Proof Nothing SAUCER," evoking a punk rock aesthetic. </think>

<think> The target description calls for text changes: "SIMPLY BULLET PROOF NOTHING SAUCER". The original image has "SIMPLY BULLET PROCD NOINISERY". To align with the target, replace the existing text with the specified words, ensuring proper capitalization and spelling. The layout and font style of the text should remain consistent with the original image's design. </think> Modify the text overlay to read "SIMPLY BULLET PROOF NOTHING SAUCER" in bold, capital letters matching the style of the original image, ensuring alignment with the target description's requirements. Final edit: Update the text to "SIMPLY BULLET PROOF NOTHING SAUCER" in a font and size consistent with the original image's design, preserving visual cohesion.

<think> The target description states the text should read "SIMPLY BULLET PROOF NOTHING SAUCER." The provided image already matches this description, with the text and the band performing on stage. Therefore, no further editing operations are required beyond what is already present in the image. </think> No further edit needed; the image already aligns with the target description of text ("SIMPLY BULLET PROOF NOTHING SAUCER") and a band performing on stage. Final Output: The existing image meets the requirements of the target description without any additional edits.

Figure 10: An illustration of our model's two-stage generation process. For the prompt 'A picture of a band performing on a stage, with the text on it: "SIMPLY", "Bullet", "Proof", "Nothing", "SAUCER"', the model's initial stage fails to correctly render some of the text. The subsequent reflect-refine stage successfully corrects the text to match the prompt, after which the process terminates automatically.

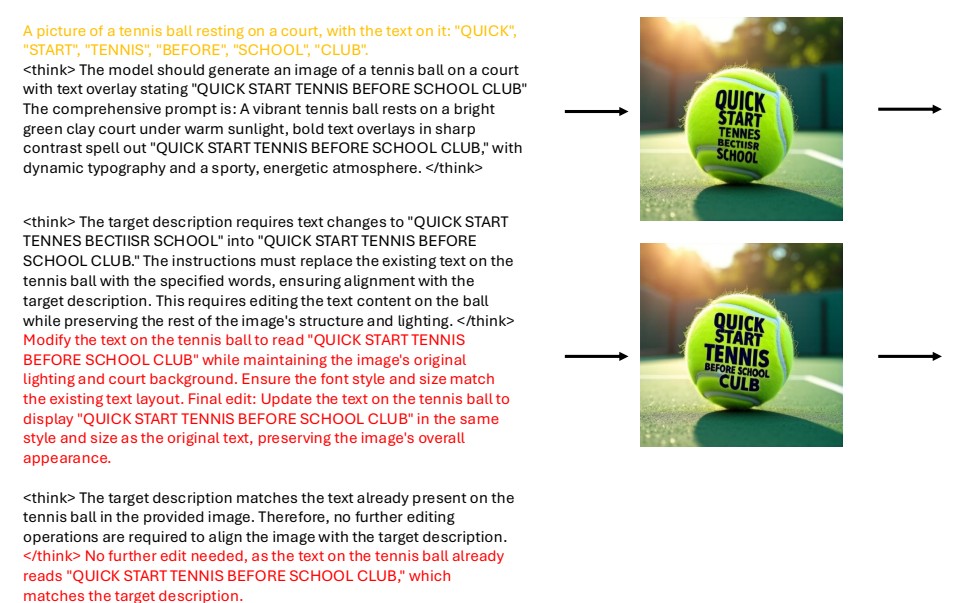

A picture of a tennis ball resting on a court, with the text on it: "QUICK", "START", "TENNIS", "BEFORE", "SCHOOL", "CLUB".

<think> The model should generate an image of a tennis ball on a court with text overlay stating "QUICK START TENNIS BEFORE SCHOOL CLUB" The comprehensive prompt is: A vibrant tennis ball rests on a bright green clay court under warm sunlight, bold text overlays in sharp contrast spell out "QUICK START TENNIS BEFORE SCHOOL CLUB," with dynamic typography and a sporty, energetic atmosphere. </think>

<think> The target description requires text changes to "QUICK START TENNES BECTIISR SCHOOL" into "QUICK START TENNIS BEFORE SCHOOL CLUB." The instructions must replace the existing text on the tennis ball with the specified words, ensuring alignment with the target description. This requires editing the text content on the ball while preserving the rest of the image's structure and lighting. </think> Modify the text on the tennis ball to read "QUICK START TENNIS BEFORE SCHOOL CLUB" while maintaining the image's original lighting and court background. Ensure the font style and size match the existing text layout. Final edit: Update the text on the tennis ball to display "QUICK START TENNIS BEFORE SCHOOL CLUB" in the same style and size as the original text, preserving the image's overall appearance.

<think> The target description matches the text already present on the tennis ball in the provided image. Therefore, no further editing operations are required to align the image with the target description. </think> No further edit needed, as the text on the tennis ball already reads "QUICK START TENNIS BEFORE SCHOOL CLUB," which matches the target description.

Figure 11: An illustration of our model's two-stage generation process, showing a limitation. For the prompt requiring specific text on a tennis ball, the initial stage fails to render all the words. The reflect-refine stage adds most of the missing text but introduces a minor spelling error ("CLUB"). Despite this inaccuracy, the model terminates the process without attempting further corrections.

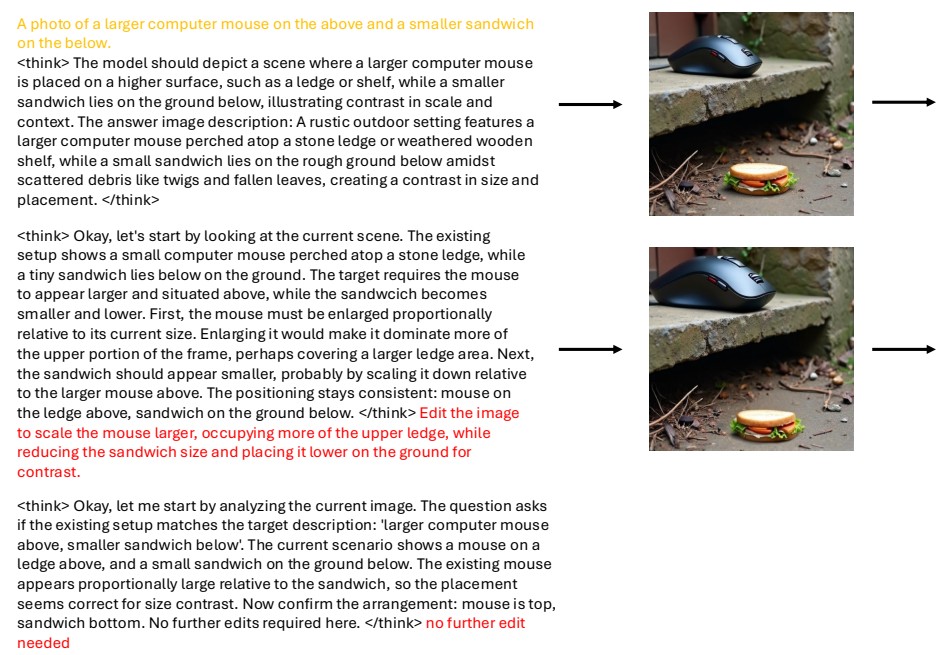

A photo of a white bench, a yellow stop sign, and a pink tie

<think> The model should depict an outdoor urban scene featuring a white bench, a yellow stop sign, and a pink necktie lying on the ground or nearby, surrounded by a natural or urban environment like grass or stone. The answer image description: A small outdoor urban setting shows a white wooden bench situated on a stone-paved area amidst overgrown grass and scattered debris. A yellow stop sign is positioned nearby, leaning slightly against a worn stone wall. A pink necktie lies discarded on the ground beside the bench, surrounded by scattered twigs and small rocks, evoking a sense of abandonment or forgotten objects in a rustic setting. </think>

<think> Okay, let's start by comparing the original image with the target description. The target requires a white bench, yellow stop sign, and pink tie. Looking at the existing image: 1. The bench: The current bench is white, so that matches the first requirement. 2. The sign: The current sign is yellow and shaped like a hexagon, not a stop sign. A stop sign is typically octagonal. So the sign must be changed to octagonal yellow. 3. The tie: The pink tie is already visible lying on the ground, so no change needed here. Therefore, the editing task involves replacing the hexagonal sign with an octagonal yellow stop sign, while keeping the rest unchanged. </think> Edit the sign to replace the hexagon with an octagonal yellow stop sign, while preserving the white bench and pink tie.

<think> Okay, let's start by comparing the original image to the target description. The target requires a white bench, yellow stop sign, and pink tie. Looking at the existing image: 1. The bench is already white, so that matches the first requirement. 2. The stop sign is yellow, which is correct. 3. The tie: The current object is pink, so no change needed here. Therefore, no further edits are required as all elements align perfectly with the target description. </think> no further edit needed

Figure 12: An illustration of our model's two-stage generation process. For the prompt "A photo of a white bench, a yellow stop sign, and a pink tie," the model's initial generation fails to correctly render the stop sign. The subsequent reflect-refine stage successfully generates the object as specified, and the process terminates automatically.

A photo of a larger computer mouse on the above and a smaller sandwich on the below.

<think> The model should depict a scene where a larger computer mouse is placed on a higher surface, such as a ledge or shelf, while a smaller sandwich lies on the ground below, illustrating contrast in scale and context. The answer image description: A rustic outdoor setting features a larger computer mouse perched atop a stone ledge or weathered wooden shelf, while a small sandwich lies on the rough ground below amidst scattered debris like twigs and fallen leaves, creating a contrast in size and placement. </think>

<think> Okay, let's start by looking at the current scene. The existing setup shows a small computer mouse perched atop a stone ledge, while a tiny sandwich lies below on the ground. The target requires the mouse to appear larger and situated above, while the sandwcich becomes smaller and lower. First, the mouse must be enlarged proportionally relative to its current size. Enlarging it would make it dominate more of the upper portion of the frame, perhaps covering a larger ledge area. Next, the sandwich should appear smaller, probably by scaling it down relative to the larger mouse above. The positioning stays consistent: mouse on the ledge above, sandwich on the ground below. </think> Edit the image to scale the mouse larger, occupying more of the upper ledge, while reducing the sandwich size and placing it lower on the ground for contrast.

<think> Okay, let me start by analyzing the current image. The question asks if the existing setup matches the target description: 'larger computer mouse above, smaller sandwich below'. The current scenario shows a mouse on a ledge above, and a small sandwich on the ground below. The existing mouse appears proportionally large relative to the sandwich, so the placement seems correct for size contrast. Now confirm the arrangement: mouse is top, sandwich bottom. No further edits required here. </think> no further edit needed

Figure 13: An illustration of our model's two-stage generation process. For the prompt "A photo of a larger computer mouse on the above and a smaller sandwich," the model's initial generation fails to correctly depict the specified size relationship. The subsequent reflect-refine stage adjusts the relative sizes of the objects to match the prompt, after which the process terminates.

A photo of a white cat, a green bench, a red elephant, and a purple sandwich

<think> The model should depict an outdoor scene featuring a white cat resting on the ground, a green park bench, a small red elephant statue, and a purple sandwich lying nearby, all set in a natural or urban environment like a park or garden area. The answer image description: A small outdoor setting shows a white cat lying on the grass near the base of a green park bench, while a small red elephant statue sits on a stone or concrete surface nearby. A purple sandwich lies discarded on the ground beside the cat, surrounded by scattered debris like fallen leaves and small twigs, evoking an urban park atmosphere. </think>

<think> Okay, let me start by comparing the current image to the target description. The current scene features a white cat lying on the ground, a green park bench in the background, a purple elephant toy to the right, and a small toy-like sandwich in the foreground. The target requires a white cat, green bench, red elephant, and purple sandwich. Let's break down each component: 1. White cat: The cat is already white and matches the requirement. 2. Green bench: The existing bench is green, so no change here. 3. Elephant: The current elephant is purple. Target requires red, so need to change elephant color to red. 4. Sandwich: The current object is a small toy sandwich, which appears yellowish-brown. Target requires a purple sandwich. So replace the existing object with a purple one. </think> Edit the elephant to be red and replace the foreground object with a purple sandwich. Keep the white cat, green bench, and other elements unchanged.

<think> Okay, let me start by comparing the current image to the target description. The current scene features a white cat lying on the ground in front of a green park bench. Next to the cat is a red elephant toy, and in the foreground lies a purple object resembling a sandwich. Now, the target requires a white cat, green bench, red elephant, and purple sandwich. First, confirm the cat color matches (white), which is correct. Next, check the bench color: the existing bench is green, which fits. The elephant color is red, matching the target. Now the sandwich part: the current foreground object is purple, so that's correct too. No further edits are needed here because all elements match the desired components. </think> no further edit needed

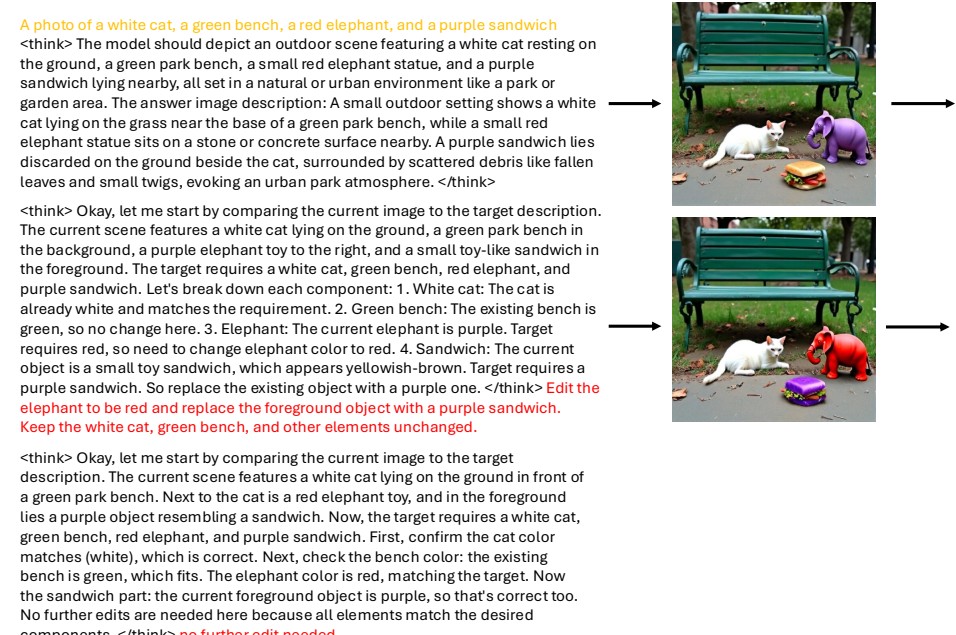

Figure 14: An illustration of our model's two-stage generation process. For the prompt "a photo of a white cat, a green bench, a red elephant, and a purple sandwich on the below," the model's initial generation renders both the elephant and the sandwich with incorrect colors. The subsequent reflect-refine stage corrects the colors of both objects, and the process terminates automatically.

A photo of two suitcases on the above and three donuts on the below.

<think> The model should depict an outdoor urban setting where two suitcases are placed on a ledge or elevated surface, while three donuts lie scattered on the ground below, surrounded by a natural or slightly worn environment for context. The answer image description: A rustic outdoor scene showcasing two vintage suitcases perched atop a stone ledge amidst a rough urban environment, while three donuts lie haphazardly on the ground below, surrounded by scattered debris like fallen leaves and small twigs, evoking a sense of abandonment and time-worn charm. </think>

<think> Okay, let's start by comparing the original image to the target description. The target requires two suitcases on the upper part and three donuts on the lower. Looking at the existing setup: Upper part: The current image shows two stacked suitcases. That matches the requirement of 'two suitcases on the above.' So no change needed here. Lower part: The current scene displays four donuts arranged in a row. The target requires 'three donuts on the below.' Therefore, the editing step would involve removing the extra donut, leaving only three visible. </think> Edit the image to remove the extra donut on the far right, leaving three donuts in the foreground arranged similarly, while preserving the two stacked suitcases on the ledge above unchanged.

<think> Okay, let me start by comparing the current image to the target description. The target requires two suitcases at the top and three donuts at the bottom. Looking at the original image: The top part has two suitcases stacked correctly, so the upper portion matches the requirement. Now the bottom: there are four donuts shown, while the target requires three. Therefore, I must remove the extra donut, leaving three remaining at the base. </think> Edit the image to remove the extra donut on the far right, leaving three donuts remaining on the bottom, while preserving the two suitcases on the top unchanged.

<think> Okay, let's see. The target requires two suitcases on the upper part and three donuts on the lower. Looking at the current image: The upper part already shows two suitcases stacked correctly on top of the other. Now check the lower part shows three donuts lying on the ground, which matches the target requirement of three donuts at the bottom. No further edits are needed here because both upper and lower sections align perfectly. Therefore, confirm no further edits required. </think> no further edit needed

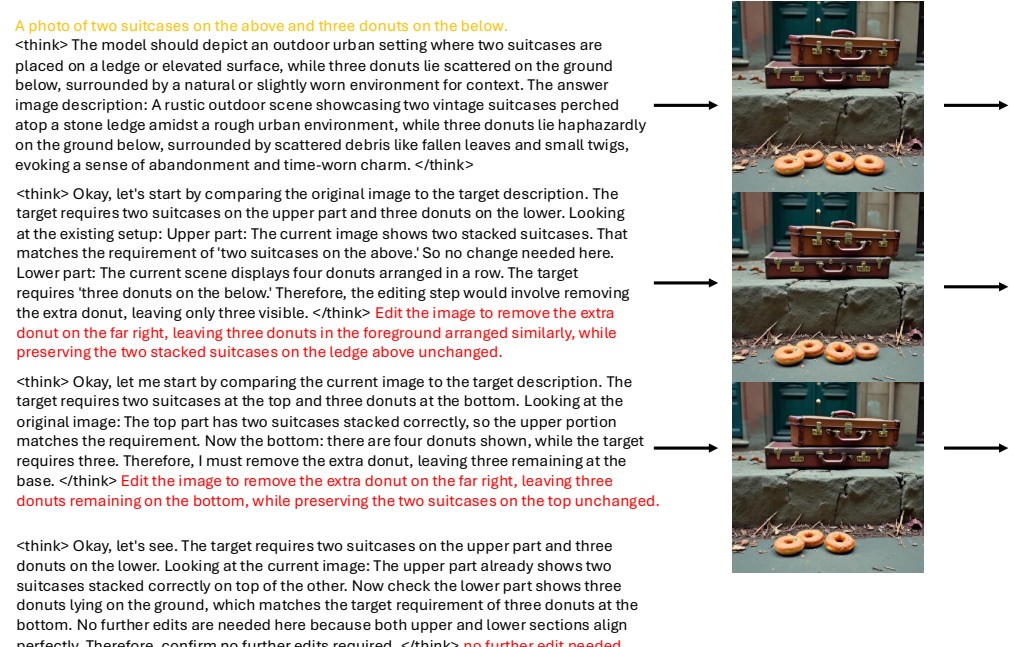

Figure 15: An illustration of our model's three-stage generation process. For the prompt "A photo of two suitcases on the above and three donuts on the below," the initial generation incorrectly produces four donuts. In the first reflect-refine stage, the model identifies the count error but fails to correct it. A second reflect-refine stage is initiated, where the model successfully edits the image to show the correct number of donuts, after which the process terminates automatically.

## A.7 Extension of R3 framework to Maze Navigation Task

To demonstrate the versatility of our proposed R3 (Reason-Reflect-Refine) framework beyond text-to-image generation, we apply it to the complex task of Maze Navigation. In this task, the model is given an initial image of a maze and is required to plot a solution path from the entrance to the exit. Generating the entire path in a single forward pass is challenging for complex mazes; therefore, the iterative nature of our R3 framework is particularly well-suited for decomposing this problem into a sequence of manageable sub-steps.

Our training strategy consists of two stages. First, we perform Supervised Fine-Tuning (SFT) on a dataset of random trajectories within various mazes. This initial stage equips the model with the fundamental capability of state transition—that is, generating the subsequent maze image based on its current state and a given movement instruction. Subsequently, we employ Reinforcement Learning (RL) to optimize the framework, enabling the model to learn an effective policy for generating a sequence of movement instructions that successfully navigates the maze from start to finish.

You are a maze navigation agent. Given current maze state image, you should:
1. Understand the maze elements: - Yellow star: starting point - Red flag: destination - Blue circle: current position - Black lines: currently exploring path - Gray lines: explored dead-end paths
2. Follow the navigation rules: - Move to the next intersection/junction in each step - When a path leads to a dead end, mark it gray, and return to the previous intersection/junction
3. Your task: Given the current maze state, you should first think and analyze the maze, then predict the next movement instruction. The next movement instruction (starts with <action> and ends with </action>), e.g. <action>Go up/down/left/right along the maze path.</action>, and output "Reached the destination\n<|termination|>" when reaching the red flag destination"'

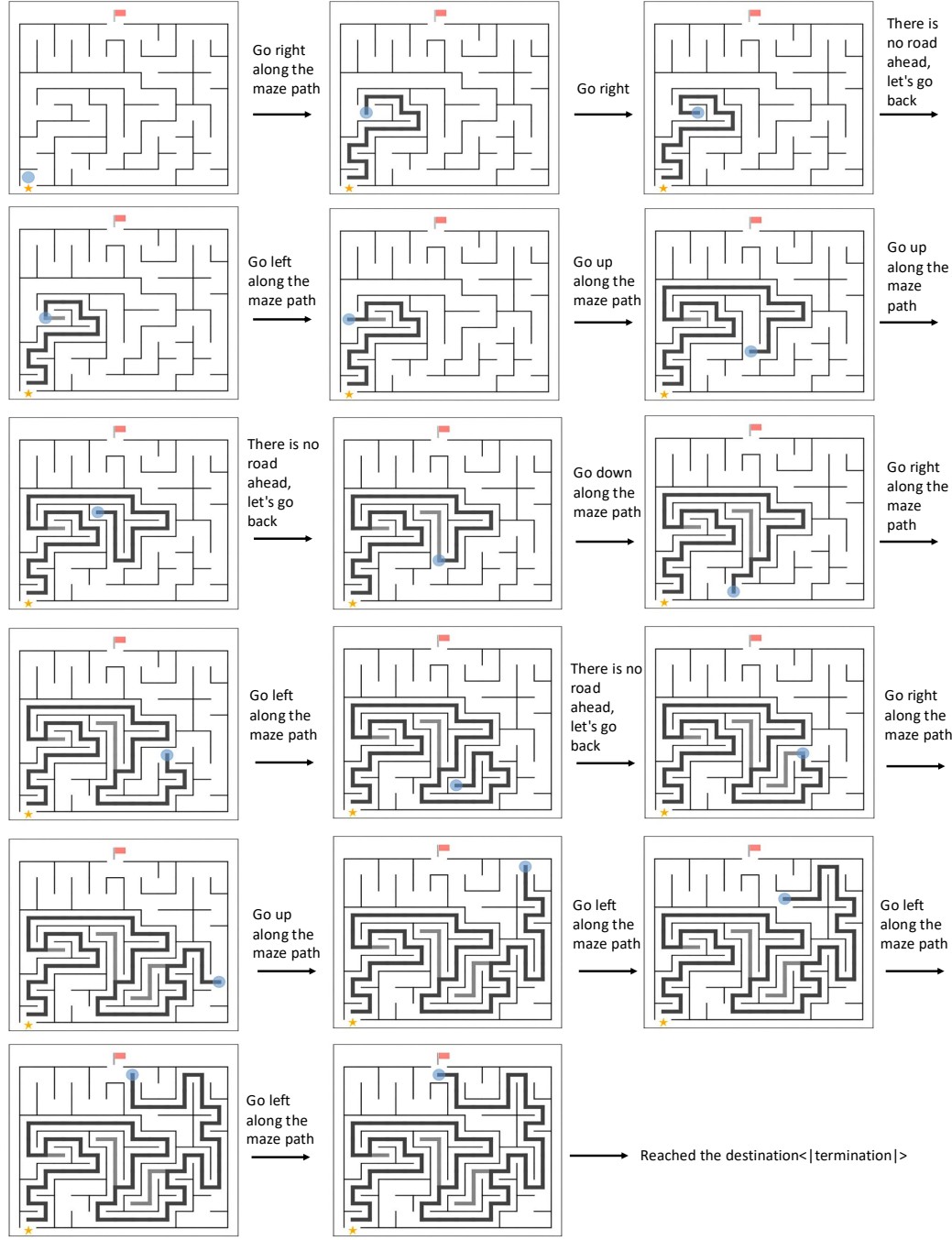

Figure 16: An illustration of maze navigation results. The model is tested on a 12x10 maze, where it demonstrates the ability to accurately identify valid directions during the navigation process. When faced with multiple paths or dead ends, the model can effectively backtrack. Furthermore, it automatically recognizes when the destination has been reached and terminates the process accordingly.

