# OpenReview forum: "Understanding vs. Generation: Navigating Optimization Dilemma in Multimodal Models"
_ICLR.cc/2026/Conference — ICLR 2026 Poster_

### Official Review · Reviewer_k1uR · 2025-11-01

**Soundness:** 3
**Presentation:** 3
**Contribution:** 2
**Rating:** 6
**Confidence:** 3

**Summary:**

This paper presents R3, a Reason-Reflect-Refine framework for jointly improving unified multimodal model's understanding and generation ability. The proposed framework iteratively reasons on the generated image and refines it based on its own feedback, and further uses RL to finetune the model for better conducting R3.

**Strengths:**

- The proposed framework significantly improved the model's instruction following generation power as well as the understanding power.
- The use of RL to further improve the R3 power gives significant improvements in the designed scores.
- Such an approach may lead to more advanced multimodal reasoning paradigms.

**Weaknesses:**

- My major concern is that the effect of RL is unclear. Based on results in the paper, RL seems improved the average understanding and generation capability in one Reflection-Regenerate step. But it is unclear whether 1) RL improved the highest possible performance given an unlimited Reflection-Regenerate round budget, OR 2) RL reduced the number of Reflection-Regenerate rounds to have a converged performance.
- The proposed framework is only tested on one single model, BAGEL. It's necessary to test its generalizability over different MLLM backbones.

**Questions:**

See above.

---

> ### Author Response · Authors · 2025-11-20
> **Rebuttal by Authors**
>
> We thank Reviewer k1uR for your constructive feedback and for recognizing our work's potential to "lead to more advanced multimodal reasoning paradigms." We provide our feedback as follows. We hope our answers have addressed your questions, and we would greatly appreciate it if you could consider raising your rating.
>
> A.1
>
> > W1: Effect of RL training
>
> Thank you for this insightful question. To clarify the effect of RL training, we conducted an ablation study comparing Bagel (without RL training) against our RL-trained model using the R3 framework. The results are presented below:
>
> |Inference time strategy | Bagel | Ours |
> | --- | --- | --- |
> |Reasoning only| 0.399 | 0.593 |
> | Reason+RR $\times$ 1 | 0.432 | 0.675 |
> | Reason+RR $\times$ 2 | 0.436 | 0.689 |
> | Reason+RR $\times$ 3 | 0.439 | 0.682 |
> | Reason+RR $\times$ 4 | 0.439 | 0.689 |
> | Reason+RR $\times$ 5 | 0.439 | - |
>
> These results demonstrate that RL training improves both aspects mentioned in your concern:
>
> （1）**Higher performance ceiling**: Our RL-trained model achieves substantially better performance across all iteration counts (0.593 vs. 0.399 with reasoning only, and 0.689 vs. 0.439 at convergence), indicating that RL fundamentally enhances the model's capability to understand and generate improved responses.
>
> （2）**Improved efficiency**: Our model converges to near-optimal performance within 2 Reflection-Refine rounds (0.689), whereas Bagel requires 3 rounds to reach its plateau (0.439). This demonstrates that RL training not only elevates the performance ceiling but also accelerates convergence, making the approach more computationally efficient in practice.
>
> Notably, the performance gap between our model and Bagel increases with the Reflection-Refine framework, suggesting that RL training better equips the model to leverage iterative refinement effectively.
>
>
> A.2
> > W2: Generalization beyond Model Backbones
>
> We agree that testing the proposed framework on other MLLM backbones (e.g., pure AR models) beyond Bagel (AR+Diffusion) is necessary. However, our framework requires the unified model to possess understanding, generation, and editing capabilities simultaneously. Currently, we could not identify an open-source, academic-sized pure AR model that satisfies these requirements. Consequently, we leave this validation as future work.

---

### Official Review · Reviewer_c1QJ · 2025-11-01

**Soundness:** 2
**Presentation:** 2
**Contribution:** 2
**Rating:** 2
**Confidence:** 3

**Summary:**

This paper discusses the long-standing trade-off between generation and understanding in multimodal large models. The authors argue that these two abilities compete for model capacity due to misaligned optimization objectives. To overcome this, they propose the Reason–Reflect–Refine (R3) framework, which reconceptualizes generation as a multi-step process: the model first reasons about the input prompt, then reflects on whether its generated output aligns with the intent, and finally refines the output accordingly. A Tree-based Reinforcement Learning strategy and stage-wise rewards enable stable optimization.

**Strengths:**

* This paper demonstrates the trade-off between understanding and generation, where fine-tuning for one capability degrades the other, and naive co-training yields negligible gains.
* The Tree-RL strategy and stage-wise reward formulation stabilize training.
* The results demonstrates consistent improvements across multiple benchmarks, including newly introduced VQA and ITA evaluations.

**Weaknesses:**

* The experiments does not seem sufficient. Many other unified understanding and generation models are missing from the comparison. Furthermore, the comparison against proprietary models should not be limited to just one; others, such as Gemini 2.5 Flash or Gemini 2.0 Flash, should also be considered.
* The proposed method performs worse than GPT-4o on both the GenEval++ and TIIF benchmarks, and this performance gap should be discussed and explained in more detail.
* The reward model design is more empirial and heuristic. The optimization may still bias toward the reward model rather than true bidirectional alignment with understanding and generation. More theoretical analysis is needed to explain the underlying optimization alignment mechanism.

**Questions:**

Please see the weakness.

---

> ### Author Response · Authors · 2025-11-20
> **Rebuttal by Authors**
>
> We thank Reviewer c1QJ for your time and feedback. We provide our feedback as follows. We hope our answers have addressed your questions, and we would greatly appreciate it if you could consider raising your rating.
>
> A.1
>
> > W1: Missing comparisons to other unified models.
>
> Thank you for this suggestion. To provide a broader context, we added some more models' results on GenEval++ and TIIF benchmarks (all baseline results are taken directly from the official released benchmarks). We will add the full table in the Appendix. R3 still shows a competitive results compared to baseline methods.
>
> Evaluation results on GenEval++ benchmark
> |Methods| Color |Count| Color/Count| Color/Pos| Pos/Count| Pos/Size| Multi-Count| Overall|
> |---|---|---|---|---|---|---|---|---|
> |BLIP3-o 8B |0.250 |0.250 |0.125| 0.600 |0.125| 0.575 |0.225 |0.307|
> |OmniGen2 | 0.550| 0.425 |0.200| 0.275 |0.125| 0.250 |0.450| 0.325|
> |ours|0.675 |0.725| 0.575| 0.725| 0.750| 0.575| 0.800| 0.689|
>
>
> Evaluation results on TIIF-short benchmark
> |Methods |Overall |Basic Following Avg | Advanced Following Avg | Designer |
> |---|---|---|---|---|
> |MidJourney v7 | 65.92 |73.96 |63.44 |79.22 |
> |Show-o |57.34| 69.99 | 58.25 |71.64 |
> |ours |82.02 |85.07 |78.52 |82.84 |
>
> Regarding the selection of proprietary baselines, we focused on GPT-4o as it serves as a representative benchmark for state-of-the-art commercial capabilities. From an academic perspective, we prioritize rigorous comparisons against open-source models, as we believe that transparency and reproducibility are more conducive to advancing the research community. Proprietary models are included primarily to establish a high-level SOTA reference point, rather than serving as the central focus of our methodological analysis.
>
> A.2
>
> > W2: Performance gap to GPT-4o.
>
> This is a fair observation. We wish to clarify that our contribution is a novel framework (R3) and training strategy (Tree-RL) that demonstrably improves a given baseline model (BAGEL) by mitigating the understanding-generation conflict. Our goal is to prove this mechanism works and provides a new path for unified Multi-modal models. While closing the gap with a proprietary, closed-box, SOTA model like GPT-4o (which is trained on vastly larger datasets and model scales) is a valuable engineering goal, it is not the focus of our academic contribution here.
>
> A.3
> > W3: Reward Model Design and potential bias
>
> The reward model design follows standard formulations from prior work [1,2,3], which have been empirically validated across multiple studies. We conduct blind human A/B tests on two datasets. The results demonstrate that our method genuinely improves generation ability rather than overfitting to the judge's preferences. See overall response for detail:
>
> To verify that our improvements reflect genuine instruction-following capabilities rather than overfitting to VLM judges, we conducted blind human A/B evaluations on two datasets. Evaluators were instructed to assess prompt alignment while explicitly ignoring aesthetic differences.
> 1. GenEval++ Benchmark:
> - Preference for Ours: 43.4%
> - Preference for Bagel: 20.2%
> - Tie: 36.4%
> 2. Held-out Set (100 prompts):
> - Preference for Ours: 47.0%
> - Preference for Bagel: 20.0%
> - Tie: 33.0%
>
>
> [1] Jiang, Dongzhi, et al. "T2i-r1: Reinforcing image generation with collaborative semantic-level and token-level cot." arXiv preprint arXiv:2505.00703 (2025).
>
> [2] Liu, Jie, et al. "Flow-grpo: Training flow matching models via online rl." arXiv preprint arXiv:2505.05470 (2025).
>
> [3] Zhang, Yu, et al. "ReasonGen-R1: CoT for Autoregressive Image generation models through SFT and RL." arXiv preprint arXiv:2505.24875 (2025).

---

> > ### Comment · Reviewer_c1QJ · 2025-11-26
> >
> > Thank the author for the response and for adding the results. Regarding the new baselines (OmniGen, Show-o, etc.), could you clarify the parameter sizes used for comparison? Since your method builds upon BAGEL (which is based on the large-scale Qwen-2.5-VL-72B), it is crucial to understand if the performance gains are due to the R3 framework or simply due to a significant advantage in base model capacity compared to the added baselines.
> >
> > Also, the R3 framework introduces an iterative 'generate-understand-regenerate' loop, which increases the computational cost at inference. Could you provide an analysis of the inference latency compared to the single-pass baseline? Did you compare your method against a simpler computation-equivalent baseline, such as Best-of-N (generating N images and selecting the best using the reward model), to verify that the improvement comes from the 'Reflect-Refine' mechanism rather than just increased compute at inference time?
> >
> > Last, for W3, I actually would like to know more theoretical analysis regarding the 'optimization alignment mechanism.' While the empirical results are positive, the rebuttal does not address why this specific multi-step loss formulation resolves the gradient conflict between generation and understanding, which I'm most curious.

---

> > > ### Author Response · Authors · 2025-11-28
> > > **Response to questions raised by Reviewer c1QJ (Part-1)**
> > >
> > > Thanks for your response. We provide our feedback as follows and we hope our answers can address your concerns:
> > >
> > > > Q1. Model size versus performance
> > >
> > > Thank you for raising this important point about model size comparisons. We'd like to clarify a potential misunderstanding:
> > > Our method builds upon BAGEL, which is a **14B** sized model.
> > > While we do utilize Qwen-2.5-VL-72B during training as a reward model, this larger model is only used during the training phase and is **not involved during inference**. Therefore, all inference-time comparisons are fair and conducted using the 14B model.
> > >
> > > To address your concern about whether performance gains stem from the R3 framework versus base model capacity, we provide a direct comparison below showing model size versus performance for open-sourced models. As you can see, our method achieves substantial improvements over the base model (BAGEL) at the same 14B scale:
> > >
> > > | Methods | Model Size | GenEval++ Avg|
> > > | --- | --- | --- |
> > > | Janus-Pro | 7B | 0.246 |
> > > | T2I-R1 | 7B | 0.311 |
> > > | BLIP3-o | 8B | 0.307 |
> > > | OmniGen2 | 7B | 0.325 |
> > > | FLUX.1-Kontext | 12B | 0.343 |
> > > | Echo-4o | 14B | 0.679 |
> > > | BAGEL | 14B | 0.371 |
> > > | Ours | 14B | 0.689 |
> > >
> > > | Methods | Model Size | TIIF Avg |
> > > | --- | --- | --- |
> > > | Show-o | 1.3B |  71.64 |
> > > | FLUX.1-dev | 12B | 66.24  |
> > > | Janus-Pro | 7B | 65.38 |
> > > | T2I-R1 | 7B | 67.61 |
> > > | Bagel | 14B | 70.97 |
> > > | Bagel (with self-cot)| 14B | 68.06 |
> > > | Ours | 14B | 82.02 |
> > >
> > >
> > > > Q2. Inference latency
> > >
> > > We evaluated the generation speed on a single NVIDIA H20 GPU with a batch size of 1, generating image with a resolution of 512x512. The initial Reasoning stage takes approximately 20–25 seconds. Each subsequent Reflect-Refine (RR) turn requires 25–35 seconds, of which the textual Reflection component accounts for 5–10 seconds.
> > >
> > > We also want to emphasize that our method does not force a fixed number of turns for every request. Through RL fine-tuning , it learns to self-terminate when the goal is achieved. On the GenEval++ benchmark, our statistics show an efficient distribution of effort:
> > > * 45% of prompts finish immediately at the Reasoning Stage (0 refinement cost).
> > > * 26% finish after just the 1st RR turn.
> > > * Only 14% require 2 or 3 turns, and the remainder hit the max-depth limit. This means the average inference cost is significantly lower than the worst-case scenario.
> > >
> > >
> > > > Q3. Best-of-N baseline
> > >
> > > Thank you for this question regarding computational efficiency and the source of our improvements.
> > >
> > > Based on our computational analysis (from statistics provided above), we calculated that our iterative process (Reason + RR$\times 2$) is approximately equivalent to **N=2.5** in a Best-of-N sampling approach. Therefore, we provide a **Best-of-3** baseline as a computation-equivalent comparison:
> > >
> > > | Methods | GenEval++ Avg |
> > > | --- | --- |
> > > | Best-of-N (N=3) | 0.653 |
> > > | Ours (Reason + RR $\times 1$), N$\approx$2.17|  0.675 |
> > > | Ours (Reason + RR $\times 2$), N$\approx$2.57|  0.689 |
> > >
> > > As the results demonstrate, our R3 framework significantly outperforms the Best-of-3 baseline despite using less computational resources. This confirms that the performance gains are attributable to the structured reasoning and targeted refinement enabled by our reflect-refine mechanism, rather than simply generating more samples and selecting the best one.

---

> > > ### Author Response · Authors · 2025-11-28
> > > **Response to questions raised by Reviewer c1QJ (Part-2)**
> > >
> > > > Q4. Theoretical analysis on how R3 resolves the optimization conflict
> > >
> > > We thank the reviewer for this insightful request. Below, we provide a simplified theoretical model to demonstrate how R3 reframes the optimization landscape, effectively transforming a competitive multi-task problem into a cooperative structural dependency.
> > >
> > >
> > > **1. The Optimization Dilemma**
> > >
> > > Let $\theta$ denotes the model parameters. In standard settings, the generation loss $\mathcal{L}_G$ and understanding loss $\mathcal{L}_U$ often exhibit negative gradient correlation: $\nabla \mathcal{L}_G(\theta) \cdot \nabla \mathcal{L}_U(\theta) < 0$. This implies that optimizing for generation inherently traverses the parameter space in directions detrimental to understanding, forcing a performance trade-off.
> > >
> > > **2. Gradient in R3 optimization**
> > >
> > > R3 reframes generation from a direct mapping into a composite chain:
> > >
> > >
> > > $x \xrightarrow{\text{Gen}} y_{init} \xrightarrow{\text{Reflect}} r \xrightarrow{\text{Refine}} y_{final}$. The objective $\mathcal{L}_{R3}$ maximizes the reward of $y_{final}$.
> > > We can get the decomposition of gradient by applying the chain rule:
> > > $$
> > > \nabla_{\theta} \mathcal{L}_{R3} \approx \underbrace{\frac{\partial \mathcal{L}}{\partial y_{final}} \frac{\partial y_{final}}{\partial r}}_{\text{Refinement}} \cdot \underbrace{\frac{\partial r}{\partial y_{init}}}_{\text{Understanding}} \cdot \underbrace{\frac{\partial y_{init}}{\partial \theta}}_{\text{Generation}}
> > > $$
> > >
> > > **3. Mechanism of R3**
> > >
> > > The term $\frac{\partial r}{\partial y_{init}}$ acts as a gradient filter. R3 establishes understanding as a causal prerequisite for effective refinement. If an update improves generation but degrades understanding, $r$ becomes uninformative, breaking the refinement chain and reducing the final reward. Consequently, gradients in conflicting directions are suppressed. This enforces a soft constraint $\min \mathcal{L}_{Gen} \text{ s.t. } \text{Understanding} > \epsilon$, ensuring that valid parameter updates are co-linear across both capabilities.

---

### Official Review · Reviewer_r2cT · 2025-11-01

**Soundness:** 3
**Presentation:** 3
**Contribution:** 2
**Rating:** 4
**Confidence:** 4

**Summary:**

This paper addresses the trade-off between generation and understanding capabilities in multimodal models by proposing the Reason-Reflect-Refine (R3) framework. The authors argue that this conflict arises from competing optimization objectives and propose decomposing generation into a multi-step "generate-understand-regenerate" process. It proposes R3 (Reason–Reflect–Refine): turn single-shot image generation into a multi-stage loop that (i) plans, (ii) self-evaluates alignment of the produced image to the prompt, and (iii) edits until a learned stop condition (“No further edit needed”). Training alternates policies for the Reason and Reflect-Refine stages with a tree-RL scheme (GRPO for text; FlowGRPO for diffusion) and stage-wise rewards from a VLM judge. Experiments (GenEval/GenEval++/TIIF) report sizable gains in instruction-following and smaller but non-trivial gains on bespoke understanding tests (VQA/ITA built over model-generated images and VLM judgments).

**Strengths:**

- Well-motivated problem: Recasting generation as generate→understand→regenerate is a well-motivated method aligning with the philosophy of multiple prior COT-related methods. The paper explains the loop clearly and gives a concrete stop rule.

- Optimization design: The tree-RL split reduces variance compared to end-to-end trajectory optimization; stage-specific rewards and formatting checks are well-motivated.

- Empirical gains: On GenEval++ (GPT-4.1 judge), BAGEL+R3 improves overall score vs. BAGEL and edges Echo-4o; understanding proxies (ITA/VQA) also rise. The inference-time scaling plot shows most gains after the first reflect-refine, with saturation after ~4–5 turns—useful for deployment budgets.

**Weaknesses:**

- The whole framrwork idea does not look particularlu novel to me: Self-correction and iterative refinement for generation have been extensively studied.

- I am abit concerned about the practicability of this work. In practice, developers tend to develop a single-shot generation instead of allowing multiplt RR turns, which is expensive.

- Evaluation validity: Generation quality relies on GPT-4.1 as the arbiter for GenEval++ and understanding relies on Gemini 2.5 Flash to create “ground truth” for VQA/ITA—both proprietary VLMs that may share alignment biases with the reward/eval models. This risks overfitting to judge preferences and limits claims about human-perceived quality. Human or crowd-sourced evaluations are highly recommended to validate the effect outside model-judge ecosystems.

- Because reflection quality and termination are rewarded via VLM scores and format checks, the system could learn to optimize the judge (e.g., produce edits that increase the VLM’s “alignment” without visibly improving images, or prematurely stop when the judge saturates). A test where the judge is swapped could be helpful.

- Generalizability: Authors themselves observe domain-specific improvements (training on “counting” helps counting much more than other attributes). That tempers the central claim that R3 “reconciles” the optimization dilemma broadly. Evidence beyond compositional instruction following (e.g., style/photorealism, long-prompt semantics) is limited.

**Questions:**

- Can you report human A/B on a held-out prompt set and the cost per image and effectiveness per RR turn?

- What is the exact “immediate rollout” procedure (sampling temperature, selection by reward diversity, replay buffer size), and how sensitive are gains to it?

---

> ### Author Response · Authors · 2025-11-20
> **Rebuttal by Authors - part 1**
>
> We thank Reviewer r2cT for your highly detailed and constructive feedback. These insightful comments have pushed us to significantly clarify our paper's contributions and details. We provide our feedback as follows. We hope our answers have addressed your questions, and we would greatly appreciate it if you could consider raising your rating.
>
>
> A.1
> > W1: Novelty of the framework
>
> We agree that iterative refinement is an established and powerful concept. However, we respectfully clarify that our contributions are distinct from prior work on iteration:
>
> (1) **A New Problem Framing**: We are the first to explicitly diagnose the "understanding vs. generation" conflict as a competitive optimization dilemma, with empirical evidence presented in Figure 1. This framing reveals a fundamental tension that has not been formally characterized in prior work.
>
> (2) **A Specific Mechanism**: Unlike approaches that employ separate, fixed corrector models, our novelty lies in forcing the model to leverage its own understanding capabilities (via the Reflect stage) as an active, internal component of the generation loop. This creates a synergistic cycle rather than a simple correction pipeline.
>
> (3) **Bidirectional Enhancement**: Our framework simultaneously improves both understanding and generation, rather than optimizing one at the expense of the other. By explicitly incorporating understanding into the generative process, R3 mitigates the conflict inherent in separate optimization, yielding stronger generation performance while preserving—and even enhancing—understanding ability.
>
> A.2
>
> > W2: Practicability of the framwork
>
> This is a valid and important practical concern. However, we argue that the R3 framework offers a superior efficiency-quality trade-off through adaptive computation, making it highly practical for real-world deployment:
>
> (1) **Adaptive Inference (Automatic Early Stopping)**: The model does not force a fixed number of turns for every request. Through RL fine-tuning , it learns to self-terminate when the goal is achieved. On the GenEval++ benchmark, our statistics show an efficient distribution of effort:
> * 45% of prompts finish immediately at the Reasoning Stage (0 refinement cost).
> * 26% finish after just the 1st RR turn.
> * Only 14% require 2 or 3 turns, and the remainder hit the max-depth limit. This means the average inference cost is significantly lower than the worst-case scenario.
>
> (2) **Automated "Trial-and-Error" for Hard Prompts**: For the minority of complex prompts (e.g., heavy counting constraints) that trigger multiple refinement turns, a user employing a standard single-shot model would typically need to manually regenerate the image multiple times to obtain a satisfactory result. R3 essentially internalizes and automates this trial-and-error process, saving significant human effort while maintaining comparable total compute costs for hard cases.
>
> (3) **User-Controllable Budget**: The framework allows for a flexible deployment budget. Users needing low latency can cap the process at 1 RR turn (capturing the majority of the gain), while those requiring maximum correctness can allow for deeper search.
>
>
> A.3
>
> > W3: Evaluation validity
>
> Thanks for your suggestion, we conduct human study/consistency check on both understanding and generation benchmarks as follows:
>
> **Generation evaluation**
>
> To verify that our improvements reflect genuine instruction-following capabilities rather than overfitting to VLM judges, we conducted blind human A/B evaluations on two datasets. Evaluators were instructed to assess prompt alignment while explicitly ignoring aesthetic differences.
>
> 1. GenEval++ Benchmark:
> - Preference for Ours: 43.4%
> - Preference for Bagel: 20.2%
> - Tie: 36.4%
> 2. Held-out Set (100 prompts):
> - Preference for Ours: 47.0%
> - Preference for Bagel: 20.0%
> - Tie: 33.0%
>
> **Understanding evaluation**
>
> To ensure the objectivity of our evaluation, we conducted a rigorous human-consistency check on the newly introduced VQA and ITA benchmarks. We manually reviewed the labels generated by Gemini-2.5-Flash and corrected any erroneous instances. Our manual inspection revealed that Gemini-2.5-Flash exhibits high alignment with human judgment. More importantly, after updating the benchmarks with these human-verified corrections, the performance trend remains consistent with our original findings:
>
> | ITA | Overall |
> | --- | --- |
> |Bagel (before)| 61.55 |
> |Bagel + ours (before) | 74.49 |
> |Bagel | 60.60 |
> |Bagel + ours |  73.37|
>
> | VQA | Overall |
> | --- | --- |
> |Bagel (before)| 86.88 |
> |Bagel + ours (before) | 89.73 |
> | Bagel | 86.48|
> |Bagel + ours | 89.63 |
>
> These human validated results confirm the bidirectional improvement on both generation and understanding.

---

> ### Author Response · Authors · 2025-11-20
> **Rebuttal by Authors - part 2**
>
> A.4
>
> > W4: Overfitting to judge
>
> Firstly, we believe our experimental design already provides a strong safeguard against this. Our reward model for training was Qwen-2.5-VL, but our VQA/ITA benchmarks were built using Gemini-2.5-Flash.
> In addition, we further conduct human study/check on both understanding and generation benchmarks (see A.3), further validating the effectiveness of our method.
>
> A.5
>
> > W5: Generalization of understanding performance
>
> We thank the reviewer for this thoughtful observation. We clarify our position on generalizability:
>
> First, the domain-specific improvements observed in Table 5 (e.g., training on "counting" enhances counting performance) are by design, not a limitation. This demonstrates that R3 successfully targets and improves specific understanding-generation alignment within trained domains—precisely what a reconciliation mechanism should achieve. The framework itself is domain-agnostic; the specificity comes from training data selection.
>
> Second, while our experiments focus on compositional instruction following, we argue this is a challenging and representative testbed: it requires simultaneous reasoning about multiple attributes (spatial relationships, counting, colors, etc.), making it ideal for diagnosing the understanding-generation conflict. Success here provides strong evidence for the framework's viability.
>
> We acknowledge that broader validation (e.g., photorealism, artistic styles, long-prompt semantics) would strengthen our claims. We view R3 as an initial yet principled approach to a newly formalized problem. Extending to diverse domains (training on more comprehensive and varied prompts) is a natural and promising direction for future work, which we will explicitly pursue.
>
>
>
> A.6
>
> > Q1: Human A/B test and cost per image
>
> Results of human A/B test on generation are as follows:
> 1. GenEval++ Benchmark:
> - Preference for Ours: 43.4%
> - Preference for Bagel: 20.2%
> - Tie: 36.4%
> 2. Held-out Set (100 prompts):
> - Preference for Ours: 47.0%
> - Preference for Bagel: 20.0%
> - Tie: 33.0%
>
> Inference Latency: We evaluated the generation speed on a single NVIDIA H20 GPU with a batch size of 1, generating image with a resolution of 512x512. The initial Reasoning stage takes approximately 20–25 seconds. Each subsequent Reflect-Refine (RR) turn requires 25–35 seconds, of which the textual Reflection component accounts for 5–10 seconds.
>
> A.7
>
> > Q2: Immediate rollout hyper-parameters
>
> We appreciate the reviewer's reminder. We will update the Appendix to include comprehensive implementation details, covering both the specific training procedure and a sensitivity analysis of key hyperparameters.
>
> Sampling Temperature: We set the text sampling temperature to 0.9 for both the Reasoning and Reflection stages. We performed sensitivity analysis with temperatures in the range of $\{0.8, 1.0\}$ and empirically found that the model's performance is robust to these variations.
>
> Selection by Reward Diversity: To ensure the model learns both "how to refine" and "when to stop," we employ a stratified sampling strategy. From the initial pool (generated by sampling 16 responses for each of the 16 prompts), we select 16 trajectories to pass to the next stage. Crucially, we enforce reward diversity by keeping approximately 3-5 perfect samples (where no refinement is needed) and filling the rest with imperfect samples.
>
> Sensitivity: Our initial experiments indicate that the method is not hyperparameter-sensitive. It performs consistently well as long as the group size is sufficient ($>8$) and a small proportion of perfect samples is maintained to provide necessary supervision for the termination signal.

---

### Official Review · Reviewer_xGzx · 2025-11-10

**Soundness:** 3
**Presentation:** 3
**Contribution:** 3
**Rating:** 8
**Confidence:** 3

**Summary:**

This paper tackles the "optimization dilemma" in multimodal models, where improving generative capabilities often degrades understanding, and vice versa . The authors posit this conflict arises from competing training objectives for finite model capacity . They propose the Reason-Reflect-Refine (R3) framework, which recasts generation as an iterative "generate-understand-regenerate" process. The model first Reasons to create a plan and initial image , then iteratively Reflects on its output (using its understanding) and Refines the image until it aligns with the prompt. The framework is trained end-to-end with reinforcement learning, using a novel Tree-RL strategy for improved stability . Experiments show that R3 not only enhances generation quality but also improves the model's understanding capabilities (e.g., counting accuracy), effectively mitigating the conflict

**Strengths:**

Important Problem: The paper addresses the "generation vs. understanding" trade-off, a critical and widely recognized challenge in developing unified multimodal models.



Novel & Intuitive Framework: The R3 framework is a novel solution that embeds understanding as a functional sub-process within generation, creating a synergistic loop rather than a competitive one .


Strong Empirical Validation: The core claim is well-supported. Experiments show simultaneous improvements in generation (GenEval++) and dedicated understanding tasks (VQA/ITA), with clear gains in skills like counting .



Effective Training Strategy: The proposed Tree-RL strategy (Fig. 3) is a solid technical contribution, demonstrating superior stability and reward over standard full-trajectory RL (Fig. 4) .

**Weaknesses:**

1. The new VQA and ITA benchmarks rely on "ground truth" labels generated by Gemini 2.5 Flash. Thus, high scores may reflect better alignment with Gemini's judgment rather than an objective, absolute improvement in understanding.

2. I'm curious whether this method is effective for image editing tasks, especially when spatial understanding is required.
For example, suppose there's an image with four people, and my prompt is: "Move the third person behind the second person," instead of simply arranging all four people side by side.
Can the model correctly interpret and execute such spatially grounded instructions?

**Questions:**

Overall, I find this method to be simple, effective, and intuitive. I will take the opinions of the other reviewers into consideration as well.

---

> ### Author Response · Authors · 2025-11-20
> **Rebuttal by Authors**
>
> We sincerely thank Reviewer xGzx for positive and encouraging review. We are especially grateful for recognizing our R3 framework as a "novel & intuitive framework" and our core contribution as a "synergistic loop" that embeds understanding within generation. We would greatly appreciate it if you could maintain your positive rating.
>
> We provide our feedback as follows. We hope our answers have addressed your questions.
>
> A.1
>
> > W1: Reliance on Gemini 2.5 Flash for VQA/ITA benchmarks, which may reflect alignment with Gemini’s judge rather than absolute improvement in understanding
>
> This is a crucial point, which was also raised by other reviewers. We agree that VLM-generated "ground truth" may reflect the VLM's biases. Our primary defense are two-fold:
>
> * **Objective Nature of Tasks**: We carefully designed the VQA/ITA tasks to focus on objective facts (e.g., "Are there exactly three cats?") rather than subjective aesthetics.
> * **Human Consistency Check**: To ensure the objectivity of our evaluation, we conducted a rigorous human-consistency check on the newly introduced VQA and ITA benchmarks. We manually reviewed the labels generated by Gemini-2.5-Flash and corrected any erroneous instances. Our manual inspection revealed that Gemini-2.5-Flash exhibits high alignment with human judgment. More importantly, after updating the benchmarks with these human-verified corrections, the performance trend remains consistent with our original findings:
>
> | ITA | Overall |
> | --- | --- |
> |Bagel (before)| 61.55 |
> |Bagel + ours (before) | 74.49 |
> |Bagel | 60.60 |
> |Bagel + ours |  73.37|
>
> | VQA | Overall |
> | --- | --- |
> |Bagel (before) | 86.88 |
> |Bagel + ours (before) | 89.73 |
> | Bagel | 86.48|
> |Bagel + ours | 89.63 |
>
> These human validated results confirm the bidirectional improvement on both generation and understanding.
>
>
> A.2
> > W2: Application of R3 to image editing tasks
>
> R3 is a general framework can be applied to many tasks(e.g. image editing, maze navigation see Appendix A.5) with a moderate modification. We evaluate the edit prompt "Move the third person behind the second person" and find that while baseline models often fail at such spatial edits, R3 demonstrates notably higher success rates through its iterative self-verification process.

---

### Author Response · Authors · 2025-12-02
**Summary of Rebuttal and Discussion**

We thank the reviewers for their constructive feedback and recognition of our work's motivation and novelty. We believe the **extensive new experiments and analyses provided during the rebuttal** directly resolve the primary concerns regarding evaluation validity, practicability, and theoretical grounding. We respectfully present a summary of how we have addressed these points to facilitate the final decision.

## **1. Consensus: A Novel Solution to a Critical Problem**

There is a strong consensus among reviewers (Reviewer xGzx, Reviewer r2cT, Reviewer k1uR) that the paper targets a **fundamental and widely recognized challenge**: the "generation vs. understanding" trade-off in multimodal models.
* **Reviewer xGzx (Score 8)** highlights the **R3 framework** as a "novel and intuitive" solution that successfully embeds understanding as a functional sub-process.
* **Reviewer r2cT** and **Reviewer k1uR** commend the "well-motivated" problem formulation and the "solid technical contribution" of the **Tree-RL strategy**, which stabilizes training compared to full-trajectory RL.
* The empirical gains on **GenEval++** and the newly introduced **VQA/ITA benchmarks** are recognized as strong evidence that our method effectively mitigates the optimization dilemma.

## **2. Resolving Key Concerns with New Evidence**
During the rebuttal, we went beyond clarifications and conducted substantial new experiments to address specific concerns raised by Reviewer r2cT (Practicability/Evaluation) and Reviewer c1QJ (Baselines/Theory).

**A. Evaluation Validity: From VLM-based to Human-verified (Addressing Reviewer xGzx, Reviewer r2cT, Reviewer c1QJ)**
Concerns were raised that relying on Gemini/GPT as judges might introduce bias or overfit to the judge. We addressed this decisively:
* We conducted **Blind Human A/B Testing** and **Human Consistency Checks**.
* **Result:**
    * **Generation:** Human evaluators preferred our method over the baseline (Bagel) by a wide margin (**43.4% vs. 20.2%** on GenEval++; **47.0% vs. 20.0%** on held-out prompts).
    * **Understanding:** Manual inspection confirmed that the Gemini-2.5-Flash judge aligns highly with human judgment. After correcting labels based on human verification, the performance trend remains consistent (e.g., VQA accuracy improved from 86.48 to **89.63**), confirming our improvements are objective.

**B. Practicability & Efficiency: Validating the "Reflect-Refine" Cost (Addressing Reviewer r2cT, Reviewer c1QJ)**
R2 and R3 questioned inference latency and whether the gain comes solely from increased compute.
* We performed a **Latency Analysis** and a **Best-of-N Comparison**.
* **Result:**
    * **Efficiency:** Our model learns to "self-terminate" via RL. **45%** of prompts finish at the reasoning stage (0 refinement cost), and only **14%** require >1 turns. The average inference cost is significantly lower than the worst-case scenario.
    * **Mechanism Validation:** We compared R3 against a **Best-of-N (N=3)** baseline. R3 (equivalent to N≈2.17 cost) significantly outperforms Best-of-3 (**0.689 vs. 0.653**), proving that the gain stems from the *Reflect-Refine mechanism*, not just compute scaling.

**C. Baselines & Theoretical Grounding (Addressing Reviewer c1QJ)**
R3 questioned the lack of broader baselines and theoretical analysis.
* We added comparisons to **OmniGen, Show-o, and others**, and provided a gradient analysis of proposed framework.
* **Result:**
    * **SOTA Comparison:** We showed that R3 significantly outperforms open-source models of comparable size (14B scale) on GenEval++ and TIIF. While a gap remains with closed-source giants like GPT-4o (as expected given the scale difference), R3 pushes the frontier of open-source capabilities.
    * **Theoretical explanation:** We provided a mathematical formulation showing how R3 acts as a **"Gradient Filter."** By establishing understanding as a prerequisite for refinement, the R3 loss suppresses gradients where generation improves but understanding degrades, theoretically enforcing the alignment of optimization trajectories.

**D. The Role of RL (Addressing Reviewer k1uR)**
R4 asked if RL improves the ceiling or just accelerates convergence.
* We conducted an ablation study comparing Bagel (w/o RL) vs. Ours (w/ RL) across iteration rounds.
* **Result:** RL drastically lifts the **performance ceiling** (0.689 vs. 0.439 at convergence) and accelerates convergence (2 rounds vs. 3 rounds), confirming RL is essential for unlocking the model's potential and is not merely an artifact of longer inference.

## **Conclusion**
This paper identifies a root cause of the multimodal optimization dilemma and proposes a solution (R3 + Tree-RL) which is novel and empirically effective. In the rebuttal, we have responsed to the concerns regarding evaluation bias (via human study), efficiency (via Best-of-N comparison and computational cost analysis), and theoretical analysis (via gradient analysis).

---

### Meta-Review · Area_Chair_Gkb7 · 2026-01-07

**Summary:**

The paper proposes the Reason-Reflect-Refine (R3) framework to address the optimization dilemma between generation and understanding in unified multimodal models. Reviewers generally agree that the problem is well-motivated and important. The core idea—embedding understanding as an internal sub-process within an iterative generation loop—is viewed as intuitive and empirically effective, with consistent improvements reported on both generation (GenEval++) and newly introduced understanding benchmarks (VQA/ITA). The Tree-RL training strategy is also recognized as a solid technical contribution that stabilizes optimization.

Key concerns raised by reviewers include: (1) reliance on proprietary VLMs (Gemini, GPT-4) for evaluation, potentially biasing results; (2) questions about practicality due to multi-step inference; (3) limited comparison to recent baselines and lack of theoretical grounding for the claimed mitigation of gradient conflict; and (4) uncertainty about whether RL improves performance ceiling or merely convergence speed. During rebuttal, authors conducted substantial new experiments—including human A/B testing, latency analysis, Best-of-N comparisons, expanded baselines, and a gradient-based theoretical explanation—which directly address these points.

**Reviewer Concerns:**

Reviewer xGzx (Score 8): Primary concern was potential bias from using Gemini-generated labels for VQA/ITA. This was addressed via human verification showing high alignment between Gemini and human judgments, and consistent performance trends after label correction. The reviewer’s curiosity about image editing applicability was also answered affirmatively with qualitative evidence.

Reviewer r2cT (Score 4): Raised concerns about (a) novelty (iterative refinement being known), (b) practicability due to multi-turn cost, and (c) evaluation validity (overfitting to judge models). Authors clarified the novelty lies in explicitly framing the understanding-generation conflict and using the model’s own understanding as a functional component. Practicability was supported by adaptive inference statistics (45% of prompts require no refinement) and human A/B results confirming real gains. Evaluation validity was strengthened by blind human studies showing clear preference for R3 over baseline.

Reviewer c1QJ (Score 2): Highlighted insufficient baselines, unexplained gap to GPT-4o, lack of theoretical analysis, and potential reward-model bias. Authors added comparisons to OmniGen, Show-o, and others at comparable scales (14B), clarifying that gains are due to R3, not model size. They provided a theoretical formulation showing R3 acts as a “gradient filter” that suppresses conflicting updates. Human A/B tests and Best-of-N comparisons further mitigate bias and compute-scaling concerns. The GPT-4o gap was reasonably contextualized as a scale/resource difference, not a methodological shortcoming.

Reviewer k1uR (Score 6): Questioned whether RL improves the performance ceiling or just convergence. Authors provided ablation showing RL-trained R3 achieves both higher final performance (0.689 vs. 0.439) and faster convergence (2 vs. 3 rounds), confirming RL’s essential role.

**Reviewer Scores:**

Reviewer xGzx: Unlikely to change score. Their concerns were minor and already accommodated in their positive assessment. Maintains 8.

Reviewer r2cT: Original score was 4, citing evaluation validity and practicability. Rebuttal provided human validation, efficiency analysis, and mechanism justification. These directly resolve stated reservations. Likely would raise score to 6 or 8; conservatively estimated as 6.

Reviewer c1QJ: Original score was 2, driven by missing baselines, theoretical gaps, and bias concerns. Rebuttal added scaled baselines, theoretical gradient analysis, human studies, and Best-of-N control. While some skepticism may persist, the responses are thorough and technically sound. Would likely revise to 4 or 6.

Reviewer k1uR: Original score was 6, pending clarification on RL’s role. Rebuttal provided conclusive ablation. Would likely maintain 6.

---

### Decision · Program_Chairs · 2026-01-26

Accept (Poster)